# Expected Free Energy-based Planning as Variational Inference

**Wouter W. L. Nuijten**                                                  *w.w.l.nuijten@tue.nl*
*Eindhoven University of Technology, Eindhoven, the Netherlands*
*Lazy Dynamics B.V., the Netherlands*

**Thijs van de Laar**                                                      *t.w.v.d.laar@tue.nl*
*Eindhoven University of Technology, Eindhoven, the Netherlands*

**Bert de Vries**                                                          *bert.de.vries@tue.nl*
*Eindhoven University of Technology, Eindhoven, the Netherlands*
*Lazy Dynamics B.V., the Netherlands*

**Reviewed on OpenReview:** *https://openreview.net/forum?id=Kzm8I1oS1s*

## Abstract

Planning under uncertainty requires agents to balance goal achievement with information gathering. Active inference addresses this through the Expected Free Energy (EFE), a cost function that unifies instrumental and epistemic objectives. However, existing EFE-based methods typically employ specialized optimization procedures that are difficult to extend or analyze. In this paper, we show that EFE-based planning can be formulated as Variational Free Energy minimization on a generative model augmented with epistemic priors. Our main result demonstrates that minimizing a Variational Free Energy functional with appropriately chosen priors yields a decomposition into expected plan costs (the EFE) plus a complexity term. This formulation reinforces theoretical consistency with the Free Energy Principle by casting planning as the same inferential process that governs perception and learning. We validate our approach on three environments of increasing complexity: a deterministic T-maze, a stochastic Reactivity Maze, and a partially observable MiniGrid DoorKey-8x8 environment. The experiments demonstrate that the epistemic priors induce information-seeking behavior, that the variational formulation yields policy-based inference outperforming plan-based methods under stochastic transitions, and that temporal factorization enables scalability to environments where existing tabular active inference methods cannot operate.

## 1 Introduction

Agents operating in uncertain environments face a fundamental tension: should they exploit current knowledge to achieve immediate goals, or explore to gather information that may improve future decisions? Classical approaches in reinforcement learning and optimal control typically address this through value function estimation or policy learning (Sutton & Barto, 2018; Bertsekas, 2012; Geffner, 2018), but often treat reward maximization and uncertainty reduction as separate objectives, relying on heuristics to balance exploration and exploitation.

Active inference offers an alternative, grounded in the Free Energy Principle (FEP), which casts perception, learning, and action selection as inference processes that minimize a variational bound on surprise (Friston, 2010; Parr et al., 2022). Central to this framework is the Expected Free Energy (EFE), an objective that combines instrumental (goal-directed) and epistemic (information-seeking) components (Friston et al., 2015). Minimizing EFE yields behavior that simultaneously pursues preferred outcomes and resolves uncertainty.

However, existing implementations of EFE-based planning typically employ specialized optimization proce-
dures, such as tree search or explicit enumeration over policies, that are difficult to extend or analyze using
standard tools (Friston et al., 2021; Paul et al., 2024a). This has limited both the theoretical understanding
of EFE minimization and its practical scalability.

In this work, we show that EFE-based planning can be formulated as variational optimization. Our central
result (Proposition 1) demonstrates that minimizing a Variational Free Energy (VFE) functional on a gen-
erative model augmented with epistemic priors yields a decomposition into expected plan costs (the EFE)
plus a complexity term. This formulation achieves theoretical alignment with the FEP: planning emerges
from the same variational principle that governs perception and learning. Our contributions are as follows:

- We show that minimizing a Variational Free Energy functional with appropriately chosen epistemic
  priors yields EFE-based planning as a natural consequence (Proposition 1).

- We identify the specific epistemic priors that induce risk-minimizing, ambiguity-reducing, and
  novelty-seeking behavior.

- We provide empirical validation on a deterministic T-maze, a stochastic Reactivity Maze, and a
  partially observable MiniGrid DoorKey-8x8 environment, demonstrating that the epistemic priors
  are sufficient for information-seeking behavior, that the variational formulation yields policy-based
  inference that outperforms plan-based methods under stochastic transitions, and that temporal
  factorization enables scalability to standard reinforcement learning benchmarks.

The results demonstrate that the epistemic priors from Proposition 1 are necessary and sufficient for inducing
the information-gathering behavior that distinguishes active inference from standard planning-as-inference
methods.

The remainder of this paper is organized as follows. Section 2 defines the planning problem and introduces
the Expected Free Energy as a cost function for plan evaluation. Section 3 reviews prior work on EFE
minimization and planning as inference. Our central contribution is presented in Section 4. Section 5
provides empirical validation on the T-maze, Reactivity Maze, and MiniGrid DoorKey-8x8, and Section 6
discusses implications for policy inference and future directions.

## 2 The Expected Free Energy Cost Function

### 2.1 The Planning Problem

In the context of sequential decision-making, planning is the process of determining an optimal course of
action to achieve specific objectives. It is important to distinguish between two related but distinct concepts
(Geffner & Bonet, 2013; Bertsekas, 2012; LaValle, 2006):

- A policy $\pi(u_t|x_{t-1})$ is a distribution over actions conditioned on states. Under the Markov assump-
  tion, the current state is a sufficient statistic for action selection.

- A plan $\boldsymbol{u} = (u_1, \ldots, u_T)$ is a fixed sequence of actions determined in advance.

Formally, sequential decision problems are often described within the framework of a Markov Decision Process
(MDP), defined by the tuple

$$(\mathcal{X}, \mathcal{A}, p(x_0), p(x_{t+1}|x_t, u_t), R(x_t, u_t), T) \,,$$

where $\mathcal{X}$ is the state space, $\mathcal{A}$ is the action space, $p(x_0)$ is the initial state distribution, $p(x_{t+1}|x_t, u_t)$
represents the transition dynamics, $R(x_t, u_t)$ is the reward function, and $T$ is the planning horizon.

The classical planning objective is to find a policy $\pi$ that maximizes expected cumulative reward:

$$\pi^* = \arg\max_\pi \mathbb{E}_\pi \left[ \sum_{t=1}^{T} R(x_t, u_t) \right] . \tag{1}$$

While this formulation is standard in reinforcement learning and optimal control (Sutton & Barto, 2018, Equation 3.13), it does not naturally accommodate epistemic objectives such as uncertainty reduction or information-seeking behavior.

## 2.2 The Generative Model

Consider an agent described by a generative model $p(\boldsymbol{y}, \boldsymbol{x}, \boldsymbol{u}, \theta)$.[1] In this paper, we are only concerned with planning, so we will assume that the model predicts a sequence of future observations. A typical example of this model would be a rollout of a state space model, for instance

$$p(\boldsymbol{y}, \boldsymbol{x}, \boldsymbol{u}, \theta) = p(x_0)p(\theta) \underbrace{\prod_{t=1}^{T} p(y_t|x_t, \theta)p(x_t|x_{t-1}, u_t, \theta)p(u_t)}_{\text{rollout to the future}}, \tag{2}$$

where $t = 0$ denotes the current time and $p(x_0)$ is the prior over the current state. In this model, $\boldsymbol{y}$ denotes the sequence of future observations, $\boldsymbol{x}$ represents the (latent) states, $\theta$ contains the model parameters, and $\boldsymbol{u}$ refers to the plan, i.e., a sequence of future actions (controls). Because all of these variables are defined as part of a model rollout into the future, they are all treated as unobserved variables. Since (2) is designed to predict how the future is expected to unfold, we refer to it as the generative model.

In model (2), the prior distribution $p(\boldsymbol{u})$ constrains allowable plans.

## 2.3 Planning-as-Inference

An alternative to reward maximization replaces the reward function with a *preference distribution* $\hat{p}(\boldsymbol{x})$ over desired future states (Levine, 2018). The preference distribution encodes the desired states, and can be understood as proportional to an exponentiated reward. Given the generative model (2) augmented with a preference distribution $\hat{p}(\boldsymbol{x})$, the planning objective becomes inferring $q(\boldsymbol{u})$, a posterior over plans encoding beliefs about which action sequences would efficiently guide the agent to these preferred states.[2]

This perspective is called Planning-as-Inference (PAI) (Attias, 2003; Toussaint & Storkey, 2006): any procedure that employs (variational) Bayesian inference on the generative model to obtain $q(\boldsymbol{u})$, or a joint posterior from which plans or policies can be derived. The result is a distribution rather than a point estimate, naturally accommodating uncertainty in the planning process.

## 2.4 The Expected Free Energy

Having established the generative model and the planning-as-inference formulation, we now turn to how candidate plans are evaluated. In the active inference literature, plans are evaluated by a cost function $G(\boldsymbol{u})$, known as the Expected Free Energy (EFE) (Da Costa et al., 2020), which is defined as

$$G(\boldsymbol{u}) = \underbrace{\mathbb{E}_q \left[ \log \frac{q(\boldsymbol{x}|\boldsymbol{u})}{\hat{p}(\boldsymbol{x})} \right]}_{\text{risk}} + \underbrace{\mathbb{E}_q \left[ \log \frac{1}{q(\boldsymbol{y}|\boldsymbol{x}, \theta)} \right]}_{\text{ambiguity}} - \underbrace{\mathbb{E}_q \left[ \log \frac{q(\theta|\boldsymbol{y}, \boldsymbol{x})}{q(\theta|\boldsymbol{x})} \right]}_{\text{novelty}}, \tag{3}$$

$$\underbrace{\hphantom{\mathbb{E}_q \left[ \log \frac{1}{q(\boldsymbol{y}|\boldsymbol{x}, \theta)} \right] - \mathbb{E}_q \left[ \log \frac{q(\theta|\boldsymbol{y}, \boldsymbol{x})}{q(\theta|\boldsymbol{x})} \right]}}_{\text{epistemic costs}}$$

where the expectations are with respect to $q = q(\boldsymbol{y}, \boldsymbol{x}, \theta|\boldsymbol{u})$, and $q$ is the result of a variational inference procedure (see Section 4 for more details). Plans with a lower $G(\boldsymbol{u})$ value are regarded as more favorable, i.e., a priori more likely to be selected. Active inference processes are based on the Free Energy Principle, a theory that accounts for the behavior of living systems as if $G(\boldsymbol{u})$ were their planning cost

---

[1] We use boldface to denote sequences of random variables, e.g., $\boldsymbol{x} = x_{1:T}$, while individual variables such as $x_t$ are not boldface.

[2] Since $q(\boldsymbol{u})$ is conditioned solely on priors, namely the generative model $p(\boldsymbol{y}, \boldsymbol{x}, \boldsymbol{u}, \theta)$ and a preference prior $\hat{p}(\boldsymbol{x})$, it would be more appropriate to speak about an updated prior rather than a posterior. For simplicity, in this paper, all distributions that result from inference are denoted by $q(\cdot)$ and termed posteriors.

function (Parr et al., 2022; Friston, 2010; 2019). On a more practical level, we summarize this theory by discussing the incentives behind the three components of $G(\boldsymbol{u})$.

- **Risk** refers to the KL-divergence between $q(\boldsymbol{x}|\boldsymbol{u})$, which is the distribution over the state trajectory that we expect to reach under plan $\boldsymbol{u}$, and the target state $\hat{p}(\boldsymbol{x})$. EFE minimization aligns with risk minimization.

- **Ambiguity** is the expected entropy $\mathbb{E}_{q(\boldsymbol{x},\theta|\boldsymbol{u})}\left[\mathbb{H}\left[q(\boldsymbol{y}|\boldsymbol{x},\theta)\right]\right]$ of future observations $\boldsymbol{y}$ under plan $\boldsymbol{u}$. Minimizing EFE biases inference toward state trajectories that predict a low-entropy distribution over future observations, which can be interpreted as a preference for states that lead to well-predicted observations.

- **Novelty** extends information-seeking behavior to include active parameter learning. Minimizing EFE results in plans that maximize the mutual information between observations $\boldsymbol{y}$ and parameters $\theta$ (averaged over states $\boldsymbol{x}$).

EFE minimization can be viewed as a unifying framework for planning under uncertainty, integrating principles from both decision theory and optimal control.

Having established the EFE as a cost function for plan evaluation, we now review prior approaches to EFE minimization and planning as inference before presenting our main theoretical contribution.

## 3 Related Work

### 3.1 Planning as Inference

The formulation of planning as probabilistic inference has a rich history. Classical optimal control (Bellman, 1954; 1966; Pontryagin, 2018; Bonet & Geffner, 2001) provides a mathematical framework for determining control inputs that minimize a predefined cost function, while Model Predictive Control extends this to real-time settings through receding horizon strategies (Bertsekas, 2012; Richalet et al., 1978; Cutler & Ramaker, 1979).

A significant paradigm shift involves recasting these problems as inference. Todorov (2006) demonstrated that a class of optimal control problems can be solved via inference in exponential family models, giving rise to KL control. Rawlik et al. (2012) extended this perspective to stochastic optimal control, showing that the Bellman equation can be derived from variational inference principles. When dealing with stochastic dynamics or state estimation under uncertainty, stochastic optimal control methods can be reformulated using variational inference (Kappen et al., 2012; Ito & Kashima, 2022), where intractable posteriors over states and controls are approximated by tractable variational distributions. The maximum causal entropy framework (Ziebart, 2010) provides a principled foundation for entropy-regularized control, connecting optimal control to probabilistic inference through the principle of maximum entropy. Under these formulations, the backward propagation of value functions corresponds to message passing on a factor graph (Levine, 2018).

The Planning-as-Inference (PAI) framework, introduced by Attias (2003) and extended by Toussaint & Storkey (2006), Toussaint (2009), and Solway & Botvinick (2012), formalizes this perspective: the objective is to infer action trajectories consistent with prior preferences over outcomes, enabling the use of approximate inference techniques such as variational inference and message passing. However, these PAI formulations focus on maximizing expected utility and do not explicitly incorporate epistemic value, i.e., the drive to reduce uncertainty. As a result, their applicability in partially observable environments is limited, as they lack a principled mechanism for information-seeking behavior.

A known challenge in PAI formulations is the phenomenon of optimistic inference (Levine, 2018). When the goal is encoded as a conditioning event, standard inference procedures can yield posteriors that implicitly assume favorable outcomes will occur regardless of the chosen actions. This arises because conditioning on goal achievement biases the posterior toward trajectories in which stochastic transitions occur to realize the goal, even when such realizations are unlikely under the agent's actual control. Lázaro-Gredilla et al.

(2024) provided a comprehensive analysis of this issue and proposed a correction based on the observation that proper sequential decision-making should account for the fact that the environment noise can not be manipulated. Specifically, their formulation augments the variational objective with an entropy correction, which turns the variational objective into a negative expectation of a reward function, essentially turning the inference procedure into expected reward maximization.

## 3.2 Active Inference

Active inference (Da Costa et al., 2020; 2024; Parr et al., 2022) offers a complementary perspective grounded in the Free Energy Principle. Rather than treating reward maximization and uncertainty reduction as separate objectives, active inference proposes that information gain constitutes an intrinsic objective. The framework balances exploration and exploitation through the Expected Free Energy (EFE) (Friston et al., 2015), which combines epistemic value (the drive to reduce uncertainty) with instrumental value (the drive to achieve preferred outcomes).

Existing methods for EFE minimization typically employ specialized optimization procedures rather than inference. Sophisticated Inference (Friston et al., 2021) evaluates policies through explicit tree search with recursive belief modeling; Branching Time Active Inference (Champion et al., 2022) extends this to handle counterfactual branches; and Dynamic Programming EFE (DPEFE) (Paul et al., 2024a;b) computes EFE recursively using dynamic programming principles. While these approaches offer computational strategies for EFE minimization, they rely on procedural algorithms for policy selection rather than deriving policy selection from inference. From the perspective of the Free Energy Principle, which posits that cognitive and behavioral processes emerge solely from Variational Free Energy minimization (Friston, 2010), policy selection should ideally arise through inference rather than through engineered algorithmic procedures.

## 3.3 Unifying EFE with Variational Inference

Several recent works have sought to reconcile EFE minimization with standard inference procedures. Palmieri et al. (2022) introduced a framework that unifies estimation and control via belief propagation on factor graphs. Building on this, Koudahl et al. (2023); van de Laar et al. (2024) proposed modifying the Variational Free Energy by subtracting a mutual information term when inference is performed over future segments of the factor graph, based on the Generalized Free Energy (Parr & Friston, 2019). This modification successfully allows message passing to account for both instrumental and epistemic value, yielding an interruptible inference procedure for policy evaluation.

A natural question is whether the same result can be achieved without requiring different objectives for different parts of the factor graph. This motivates the present work, which seeks a formulation where EFE-based planning emerges from a single, unified variational objective.

# 4 EFE-based Planning as Variational Inference

The central construction of this paper is the following result, which shows that EFE-based planning can be formulated as Variational Free Energy minimization without requiring modified cost functions or procedural algorithms. By augmenting the generative model with specific epistemic priors, the variational objective decomposes into expected plan costs (the EFE) plus a complexity term. The epistemic priors account for entropic contributions from conditional state, observation, and parameter beliefs, corresponding to the risk, ambiguity, and novelty terms, respectively.

**Proposition 1** (EFE-based Planning as Variational Inference). *Consider an agent with generative (predictive) model $p(\boldsymbol{y}, \boldsymbol{x}, \boldsymbol{u}, \theta)$ and prior beliefs $\hat{p}(\boldsymbol{x})$ about future desired states. We define the Variational Free*

*Energy functional $F[q]$ as*

$$F[q] \triangleq \mathbb{E}_{q(\boldsymbol{y},\boldsymbol{x},\boldsymbol{u},\theta)}\left[\log \frac{\overbrace{q(\boldsymbol{y},\boldsymbol{x},\boldsymbol{u},\theta)}^{posterior}}{\underbrace{p(\boldsymbol{y},\boldsymbol{x},\boldsymbol{u},\theta)}_{\substack{generative \\ model}} \underbrace{\hat{p}(\boldsymbol{x})}_{\substack{preference \\ prior}} \underbrace{\tilde{p}(\boldsymbol{u})\tilde{p}(\boldsymbol{x})\tilde{p}(\boldsymbol{y},\boldsymbol{x})}_{epistemic\ priors}}\right], \tag{4}$$

*where the generative model in the denominator is augmented by both a preference prior $\hat{p}(\cdot)$ and epistemic priors $\tilde{p}(\cdot)$. Assume that the variational posterior $q(\boldsymbol{y},\boldsymbol{x},\boldsymbol{u},\theta)$ factorizes as*

$$q(\boldsymbol{y},\boldsymbol{x},\boldsymbol{u},\theta) = q(\boldsymbol{y},\theta|\boldsymbol{x})q(\boldsymbol{x}|\boldsymbol{u})q(\boldsymbol{u}). \tag{5}$$

*If the epistemic priors are chosen as*

$$\tilde{p}(\boldsymbol{u}) \propto \exp(\mathbb{H}\left[q(\boldsymbol{x}|\boldsymbol{u})\right]), \tag{6a}$$
$$\tilde{p}(\boldsymbol{x}) \propto \exp(\mathbb{E}_{q(\theta|\boldsymbol{x})}\left[-\mathbb{H}\left[q(\boldsymbol{y}|\boldsymbol{x},\theta)\right]\right]), \tag{6b}$$
$$\tilde{p}(\boldsymbol{y},\boldsymbol{x}) \propto \exp(\mathbb{D}_{\mathrm{KL}}\left[q(\theta|\boldsymbol{y},\boldsymbol{x})\|q(\theta|\boldsymbol{x})\right]), \tag{6c}$$

*then $F[q]$ decomposes as*

$$F[q] = \underbrace{\mathbb{E}_{q(\boldsymbol{u})}\left[G(\boldsymbol{u})\right]}_{\substack{expected\ plan \\ costs}} + \underbrace{\mathbb{E}_{q(\boldsymbol{y},\boldsymbol{x},\boldsymbol{u},\theta)}\left[\log \frac{q(\boldsymbol{y},\boldsymbol{x},\boldsymbol{u},\theta)}{p(\boldsymbol{y},\boldsymbol{x},\boldsymbol{u},\theta)}\right]}_{complexity} + constant, \tag{7}$$

*where $G(\boldsymbol{u})$ is the Expected Free Energy as defined in* (3).

In (6), $\mathbb{H}\left[q(\boldsymbol{x}|\boldsymbol{u})\right] = -\sum_{\boldsymbol{x}} q(\boldsymbol{x}|\boldsymbol{u})\log q(\boldsymbol{x}|\boldsymbol{u})$ denotes the entropy functional, and $\mathbb{D}_{\mathrm{KL}}\left[q(\theta|\boldsymbol{y},\boldsymbol{x})\|q(\theta|\boldsymbol{x})\right] = \sum_{\theta} q(\theta|\boldsymbol{y},\boldsymbol{x})\log \frac{q(\theta|\boldsymbol{y},\boldsymbol{x})}{q(\theta|\boldsymbol{x})}$ is the Kullback-Leibler divergence.

*Proof.* The proof of (7) is given in Appendix A. $\square$

A key consequence of (7) is that minimizing $F[q]$ leads to reducing the expected plan costs $\mathbb{E}_{q(\boldsymbol{u})}\left[G(\boldsymbol{u})\right]$, while also balancing this with the drive to reduce complexity, which are the costs associated with changing beliefs.

## 4.1 Optimal Posterior over Plans

Starting from (20c) (see proof in Appendix A), we can compute the optimal $q^*(\boldsymbol{u})$ through

$$F[q] = \mathbb{E}_{q(\boldsymbol{u})}\left[\log \frac{q(\boldsymbol{u})}{p(\boldsymbol{u})} + G(\boldsymbol{u}) + \underbrace{\mathbb{E}_{q(\boldsymbol{y},\boldsymbol{x},\theta|\boldsymbol{u})}\left[\log \frac{q(\boldsymbol{y},\boldsymbol{x},\theta|\boldsymbol{u})}{p(\boldsymbol{y},\boldsymbol{x},\theta|\boldsymbol{u})}\right]}_{=C(\boldsymbol{u})\ (complexity)}\right]$$

$$= \mathbb{E}_{q(\boldsymbol{u})}\left[\log \frac{q(\boldsymbol{u})}{\exp\left(-P(\boldsymbol{u}) - G(\boldsymbol{u}) - C(\boldsymbol{u})\right)}\right], \tag{8}$$

where $P(\boldsymbol{u}) = -\log p(\boldsymbol{u})$ denotes the plan prior expressed as a cost function. Equation (8) is a Kullback-Leibler divergence that is minimized for

$$q^*(\boldsymbol{u}) = \arg\min_q F[q]$$
$$= \sigma\left(-P(\boldsymbol{u}) - G(\boldsymbol{u}) - C(\boldsymbol{u})\right), \tag{9}$$

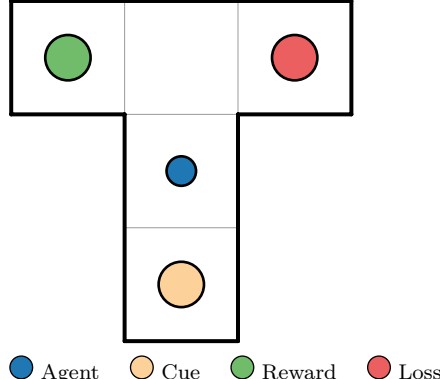

Figure 1: T-maze environment. The agent starts at the center junction and must reach the rewarding arm. Visiting the cue location (orange) provides an observation that reveals the reward location. The two arms (green and red) represent potential goal states.

where

$$\sigma(a)_k = \frac{\exp(a_k)}{\sum_{k'} \exp(a_{k'})} \tag{10}$$

is the softmax function.

Equation (9) is not new. A comparable formula for $q^*(\boldsymbol{u})$ can be found in Equation 2.1 of (Friston et al., 2021). A main contribution of this paper is to demonstrate that (9) can be obtained through variational optimization of an appropriately defined free energy functional $F[q]$. Importantly, while $q^*(\boldsymbol{u})$ is a distribution over plans (action sequences), the full joint posterior $q(\boldsymbol{y}, \boldsymbol{x}, \boldsymbol{u}, \theta)$ can be marginalized to obtain state-conditioned action beliefs $q(u_t|x_{t-1})$, effectively yielding a policy. This distinction is elaborated in Section 6.

## 5    Validation

This section validates the influence of epistemic priors on planning through three environments of increasing complexity: a deterministic T-maze, a stochastic Reactivity Maze, and a partially observable MiniGrid DoorKey-8x8.

### 5.1    Environments

We evaluate on three environments that each isolate different aspects of the planning problem.

#### 5.1.1    T-maze

The T-maze (Figure 1) is a canonical benchmark for information-seeking behavior in active inference (Friston et al., 2015). Our variant consists of five locations: a starting position, a center junction, two goal arms (left and right), and a cue location below the junction. Unlike traditional formulations where agents can transition directly between any locations, we require deterministic movement between adjacent locations only, which increases the planning horizon required to gather information. Figure 1 shows the initial state of the environment, with the reward location (green) on the left arm.

One of the two arms contains a reward; the rewarding arm is determined by a hidden context variable that remains fixed within each episode. At the cue location, the agent receives an observation that unambiguously reveals which arm contains the reward. At all other locations, observations are uninformative.

### 5.1.2   Reactivity Maze

The T-maze has deterministic transitions, so it cannot distinguish plan-based from policy-based inference. To validate this distinction, we introduce the Reactivity Maze, inspired by the reactivity environment of Lázaro-Gredilla et al. (2024, Appendix I), extended with epistemic uncertainty over multiple goal states.

The environment has two state entities. The first entity $x_t^{(1)} \in \{0, 1, 2, 3, 4, 5, 6\}$ describes the agent's location, where states 0–4 are navigation states, state 5 is an absorbing safe sink, and state 6 is an instructional cue. The second entity $x_t^{(2)} \in \{0, 1, 2, 3, 4\}$ is a reactivity knob that governs the stochasticity of transitions. There are 8 actions: actions 0–4 are navigation actions, actions 5 and 6 decrease and increase the knob, and action 7 visits the cue. A latent context parameter $\theta \in \{0, 1\}$ determines which navigation state is the optimal goal.

The knob entity has deterministic dynamics:

$$x_{t+1}^{(2)} = \begin{cases} \max(0,\ x_t^{(2)} - 1) & \text{if } a_t = 5\,, \\ \min(4,\ x_t^{(2)} + 1) & \text{if } a_t = 6\,, \\ x_t^{(2)} & \text{otherwise}\,. \end{cases} \tag{11}$$

The knob value controls the stochasticity of the location entity. For navigation actions $0 \le a_t \le 4$ at non-special locations ($x_t^{(1)} \notin \{5, 6\}$):

$$x_{t+1}^{(1)} = \begin{cases} (x_t^{(1)} + a_t) \bmod 5 & \text{with probability } x_t^{(2)}/4\,, \\ 5 \text{ (safe sink)} & \text{with probability } 1 - x_t^{(2)}/4\,. \end{cases} \tag{12}$$

Knob actions ($a_t \in \{5, 6\}$) follow the same stochastic pattern: with probability $x_t^{(2)}/4$ the agent remains at its current location, otherwise it falls to the safe sink. Action 7 deterministically moves the agent to the cue location. At the cue ($x_t^{(1)} = 6$), any action returns the agent to a uniformly random navigation state. The safe sink ($x_t^{(1)} = 5$) is absorbing.

At the cue location, the agent receives an observation that deterministically reveals $\theta$; at all other locations, observations are uninformative. Additionally, the action prior assigns a lower probability to the cue action ($p(a_t = 7) \propto 1/\epsilon$ with $\epsilon = 1.2$), making information-gathering explicitly costly: the agent must overcome this prior penalty to visit the cue. Reward is delivered only at the final timestep:

$$R_T(x_T^{(1)}, x_T^{(2)}) = \begin{cases} +1.0 & \text{if } x_T^{(1)} = \theta \text{ and } x_T^{(2)} = 4\,, \\ -1.0 & \text{if } x_T^{(1)} \neq \theta,\ x_T^{(1)} < 5 \text{ and } x_T^{(2)} = 4\,, \\ +0.33 & \text{if } x_T^{(1)} = 5\,, \\ -0.33 & \text{if } x_T^{(1)} \neq 5 \text{ and } x_T^{(2)} < 4\,. \end{cases} \tag{13}$$

The reward structure creates a tension: maximum reward ($+1.0$) requires maintaining full reactivity ($x_T^{(2)} = 4$) and reaching the correct goal ($x_T^{(1)} = \theta$), but at full reactivity the action that reaches the goal depends on the current location, requiring the agent to *react* to its state. By reducing the knob, the agent can guarantee reaching the safe sink ($+0.33$) without needing reactivity, but at the cost of a lower reward. The instructional cue resolves $\theta$ but costs a timestep and carries a prior penalty, adding a further exploration–exploitation tradeoff. Optimal behavior thus requires *both* reactive policies and epistemic learning.

### 5.1.3   MiniGrid DoorKey

To evaluate whether the variational formulation scales beyond the small state spaces of the previous experiments, we test on the DoorKey-8x8 environment from the MiniGrid suite (Chevalier-Boisvert et al., 2023). The agent operates in a $6 \times 6$ internal grid ($8 \times 8$ including grid walls) divided into two rooms by a wall with a locked door. It must find a key, pick it up, unlock the door, and navigate to the goal location. The environment is partially observable: the agent perceives only a $3 \times 3$ field of view centered on its position, reduced from the standard $7 \times 7$ configuration. This limited visibility means the agent cannot see the full room and must actively explore to locate the key and door, making the task substantially harder than the standard configuration.

### 5.2 Generative Model

The agent maintains a generative model of the form (2) for planning over a horizon of $T = 4$ timesteps. The model components are specified as follows.

- **States** $x_t \in \mathcal{X}$: The five locations, with $|\mathcal{X}| = 5$.

- **Actions** $u_t \in \mathcal{U}$: Four movement directions (up, down, left, right), with $|\mathcal{U}| = 4$. The transition model $p(x_t|x_{t-1}, u_t)$ encodes deterministic adjacency-respecting movement.

- **Observations** $y_t \in \mathcal{Y}$: Binary observations with $|\mathcal{Y}| = 2$. The observation model $p(y_t|x_t, \theta)$ yields an informative observation (deterministically revealing $\theta$) when $x_t$ is the cue location, and a uniform distribution over observation values otherwise.

- **Context parameter** $\theta \in \{\text{left}, \text{right}\}$: Indicates which arm contains the reward. The prior $p(\theta)$ reflects the agent's current belief, initialized to uniform and updated through state inference.

- **Preference prior**: The preference distribution factorizes as $\hat{p}(\boldsymbol{x}) = \hat{p}(x_T) \prod_{t=1}^{T-1} \hat{p}(x_t)$, where intermediate states have uniform preferences $\hat{p}(x_t) \propto 1$, and the terminal preference is $\hat{p}(x_T|\theta) = \sigma(\gamma \cdot \mathbb{1}[x_T = \theta])$ with $\gamma = 2$, placing high probability on reaching the arm matching the reward location.

The generative models for the Reactivity Maze and MiniGrid DoorKey follow directly from their respective environment specifications. For MiniGrid DoorKey, the context parameter $\theta$ encodes the locations of the key and door, while the state $x_t$ comprises the agent's position, orientation, and whether the key has been picked up or the door has been opened.

### 5.3 Comparison of Objectives

To isolate the contribution of epistemic priors, we define three objectives for comparison, all formulated as VFE minimization over a joint posterior $q(\boldsymbol{y}, \boldsymbol{x}, \boldsymbol{u}, \theta)$:

$$F_{\text{marginal}}[q] = \mathbb{E}_{q(\boldsymbol{y}, \boldsymbol{x}, \boldsymbol{u}, \theta)} \left[ \log \frac{q(\boldsymbol{y}, \boldsymbol{x}, \boldsymbol{u}, \theta)}{p(\boldsymbol{y}, \boldsymbol{x}, \boldsymbol{u}, \theta)\hat{p}(\boldsymbol{x})} \right] , \tag{14}$$

$$F_{\text{planning}}[q] = F_{\text{marginal}}[q] + \sum_{t=1}^{T} \left( \mathbb{H}\left[q(x_{t-1}, u_t)\right] - \mathbb{H}\left[q(x_{t-1})\right] \right) , \tag{15}$$

$$F_{\text{active}}[q] = \mathbb{E}_{q(\boldsymbol{y}, \boldsymbol{x}, \boldsymbol{u}, \theta)} \left[ \log \frac{q(\boldsymbol{y}, \boldsymbol{x}, \boldsymbol{u}, \theta)}{p(\boldsymbol{y}, \boldsymbol{x}, \boldsymbol{u}, \theta)\hat{p}(\boldsymbol{x})\tilde{p}(\boldsymbol{x})\tilde{p}(\boldsymbol{u})\tilde{p}(\boldsymbol{y}, \boldsymbol{x})} \right] . \tag{16}$$

Equation (14) is standard VFE with preference prior, consistent with KL control (Todorov, 2006; Rawlik et al., 2012, Equation 3). Equation (15) applies the entropy correction from Lázaro-Gredilla et al. (2024, Equation 4) to account for sequential decision structure and addresses the optimistic planning tendency of KL control by accounting for aleatoric uncertainty in state transitions (Levine, 2018). Equation (16) augments VFE with the full set of epistemic priors from Proposition 1. The main question is whether these epistemic priors provide additional value by inducing information-seeking behavior that anticipates the reduction of epistemic uncertainty.

### 5.4 Inference Procedure

Planning is performed by minimizing the Variational Free Energy over a joint posterior $q(\boldsymbol{y}, \boldsymbol{x}, \boldsymbol{u}, \theta) = q(\boldsymbol{y}, \theta|\boldsymbol{x})q(\boldsymbol{x}|\boldsymbol{u})q(\boldsymbol{u})$, respecting the factorization assumption in Proposition 1, using the Adam optimizer (Kingma & Ba, 2015), implemented in JAX (Bradbury et al., 2018). The variational distribution is represented as a tabular categorical distribution, with scalability implications discussed in Section 6.3. Optimization runs for 5000 iterations per planning step (see Appendix B for convergence diagnostics).

Table 1: Performance comparison on the T-maze (200 episodes). Brackets denote 95% CIs.

| Method | Success Rate | Mean Reward | Cue Visit Rate | Avg. Steps |
|---|---|---|---|---|
| $F_{\text{active}}$ (ours) | **100%** [98, 100] | $\mathbf{+1.00} \pm 0.00$ | **100%** [98, 100] | $4.0 \pm 0.0$ |
| $F_{\text{marginal}}$ | 48% [40, 55] | $-0.05 \pm 1.00$ | 0% [0, 2] | $4.0 \pm 0.0$ |
| $F_{\text{planning}}$ | 52% [45, 60] | $+0.05 \pm 1.00$ | 0% [0, 2] | $3.0 \pm 0.0$ |
| Sophisticated Inference | **100%** [98, 100] | $\mathbf{+1.00} \pm 0.00$ | **100%** [98, 100] | $4.0 \pm 0.0$ |
| Standard EFE planning | **100%** [98, 100] | $\mathbf{+1.00} \pm 0.00$ | **100%** [98, 100] | $4.0 \pm 0.0$ |

At each timestep, the agent receives an observation from the environment and updates its belief over the current state and context parameter $\theta$. Since the cue observation deterministically reveals $\theta$ and observations elsewhere reveal the current location, state inference reduces to direct belief updates. The agent then performs planning by minimizing the VFE objective over horizon $T$, and executes the action with highest marginal probability:

$$\hat{u}_{t+1} = \arg\max_{u_{t+1}} q(u_{t+1}). \tag{17}$$

This receding-horizon scheme replans at each timestep, incorporating new observations. Episodes terminate upon reaching a goal arm, with a maximum of $T$ steps.[3]

All three VFE objectives are compared on each environment, alongside Sophisticated Inference and Standard EFE planning (Friston et al., 2021) implemented using pymdp (Heins et al., 2022) where applicable.

**Reactivity Maze.** Unlike the T-maze, which uses a full tabular joint posterior that scales exponentially with the horizon $T$, the Reactivity Maze employs a temporal (Markovian) factorization:

$$q(\boldsymbol{y}, \boldsymbol{x}, \boldsymbol{u}, \theta) = q(\theta) \prod_{t=1}^{T} q(u_t|x_{t-1}) \, q(x_t|x_{t-1}, u_t, \theta) \, q(y_t|x_t, \theta), \tag{18}$$

where each factor is a variational distribution optimized through VFE minimization. This factorization scales linearly with the horizon and naturally represents reactive policies $q(u_t|x_{t-1})$ that condition actions on the current state. The inference procedure otherwise follows the T-maze setup, with 1000 optimization steps per planning step over a horizon of $T = 3$ and a maximum of 5 environment steps per episode.

**MiniGrid DoorKey.** The state and observation spaces of this environment are too large for tabular enumeration. Standard EFE planning and Sophisticated Inference, as implemented in pymdp (Heins et al., 2022), require explicit enumeration of all states and observations, and cannot operate in this setting. We therefore compare only the three VFE objectives, using the same temporal factorization as the Reactivity Maze (Equation 18). Optimization uses a planning horizon of $T = 20$, with 3000 gradient steps per planning step, a learning rate of 0.01, and a maximum of 35 environment steps per episode over 100 episodes.

### 5.5 Results

For proportions we report 95% confidence intervals; for continuous values we report $\pm 1$ standard deviation.

#### 5.5.1 T-maze

We evaluate each objective over 200 episodes with the reward location $\theta$ sampled uniformly. Table 1 summarizes the results.

All three methods with epistemic terms ($F_{\text{active}}$, Sophisticated Inference, and Standard EFE planning) achieve perfect performance, visiting the cue in every episode and always reaching the correct arm. In this deterministic setting, plan-based and policy-based methods are equally effective; the distinction between them only emerges under stochastic transitions (Section 5.5.2).

---

[3]Code is available at `https://github.com/biaslab/EpistemicPriorsGradientDescentJAX`.

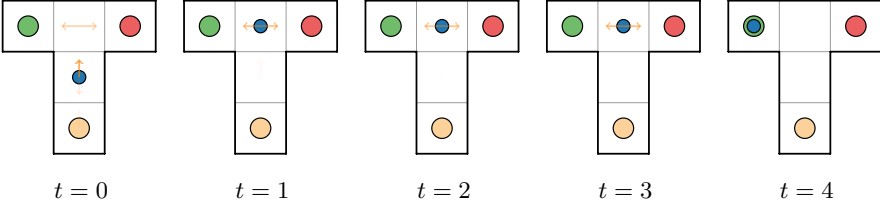

(a) $F_{\mathrm{marginal}}$: The agent proceeds toward an arm. The plan shows no preference for visiting the cue; without epistemic priors, the agent cannot anticipate the value of information.

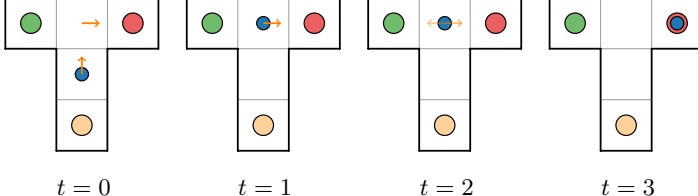

(b) $F_{\mathrm{planning}}$: Similar to marginal inference, the agent proceeds directly to an arm. The entropy correction prevents optimistic inference but does not induce information-seeking behavior.

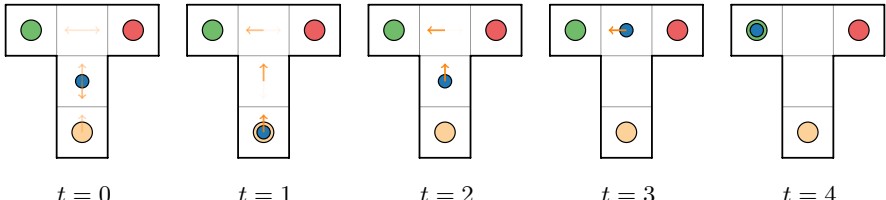

(c) $F_{\mathrm{active}}$: At $t = 0$, the plan reflects intention to visit the cue, with probability mass on the downward action. After observing the cue ($t = 1$), the agent forms a strong preference for the correct arm, visible in the concentrated arrow at subsequent times.

Figure 2: Representative trajectories with planned actions (reward in left arm). The blue circle indicates agent position. Arrows visualize the complete plan: at each location, opacity encodes the marginal probability $q(u_t|x_{t-1})$ assigned to that action at future timestep $t$. Only $F_{\mathrm{active}}$ visits the cue (orange) before committing to a goal.

Both $F_{\mathrm{marginal}}$ and $F_{\mathrm{planning}}$ achieve complementary success rates that sum to 100%, reflecting that each consistently commits to a different arm without gathering information. This asymmetry is a numerical artifact of the optimization; the key observation is that neither objective visits the cue. Without epistemic priors, both objectives yield posterior action distributions that are approximately uniform over the non-downward actions, providing no mechanism to anticipate that visiting the cue would resolve uncertainty about $\theta$.

Figure 2 illustrates representative trajectories. Each frame shows the agent's current position (blue circle) along with its complete plan: arrows at all locations indicate the marginal action probabilities $q(u_t|x_{t-1})$ the agent assigns to future timesteps, with opacity proportional to probability. This visualization reveals what the agent intends to do not only at its current location, but also at states it anticipates visiting later in the episode.

The planned actions reveal a key distinction. Under $F_{\mathrm{marginal}}$ and $F_{\mathrm{planning}}$, the agent is ambivalent at the junction: with no mechanism to value information, it proceeds directly toward an arm with roughly equal probability for left and right. In contrast, under $F_{\mathrm{active}}$, the agent visits the cue location first. After observing the cue at $t = 2$, it exhibits a strong preference for the correct arm. These results demonstrate

Table 2: Performance comparison on the Reactivity Maze (200 episodes). Brackets denote 95% CIs.

| Method | Mean Reward | Success Rate | Safe Rate | Cue Visit Rate |
|---|---|---|---|---|
| $F_{\text{active}}$ (ours) | $\mathbf{+1.00} \pm 0.00$ | $\mathbf{100\%}$ [98, 100] | $0\%$ [0, 2] | $\mathbf{100\%}$ [98, 100] |
| $F_{\text{marginal}}$ | $+0.33 \pm 0.00$ | $0\%$ [0, 2] | $100\%$ [98, 100] | $0\%$ [0, 2] |
| $F_{\text{planning}}$ | $+0.18 \pm 0.63$ | $17\%$ [12, 23] | $64\%$ [56, 70] | $0\%$ [0, 2] |
| Sophisticated Inference | $+0.96 \pm 0.16$ | $94\%$ [90, 97] | $6\%$ [3, 10] | $\mathbf{100\%}$ [98, 100] |
| Standard EFE planning | $+0.56 \pm 0.66$ | $61\%$ [54, 68] | $21\%$ [16, 27] | $66\%$ [60, 73] |

Table 3: Performance comparison on MiniGrid DoorKey-8x8 (100 episodes). Brackets denote 95% CIs. Standard EFE planning and Sophisticated Inference are excluded as they cannot handle the environment's state and observation space.

| Method | Success Rate | Mean Reward | Avg. Steps |
|---|---|---|---|
| $F_{\text{active}}$ (ours) | $\mathbf{89\%}$ [81, 94] | $\mathbf{0.85} \pm 0.30$ | $30.69 \pm 2.16$ |
| $F_{\text{marginal}}$ | $4\%$ [1, 10] | $0.04 \pm 0.19$ | $34.34 \pm 3.52$ |
| $F_{\text{planning}}$ | $\mathbf{89\%}$ [81, 94] | $\mathbf{0.86} \pm 0.30$ | $\mathbf{27.28} \pm 5.50$ |

that the epistemic priors from Proposition 1 are sufficient to induce information-seeking behavior within the variational framework.

### 5.5.2 Reactivity Maze

We evaluate all five methods over 200 episodes with $\theta$ sampled uniformly. Table 2 summarizes the results.

Only $F_{\text{active}}$ and Sophisticated Inference consistently seek information (100% cue visit rate) and achieve near-optimal reward. $F_{\text{active}}$ achieves perfect performance (100% success rate), outperforming Sophisticated Inference (94%) while relying on variational optimization rather than recursive belief modeling.

$F_{\text{marginal}}$ never seeks information, converging to the safe sink in every episode for a guaranteed +0.33 reward. $F_{\text{planning}}$ similarly avoids the cue and defaults to the safe sink in 64% of episodes. Standard EFE planning visits the cue in only 66% of episodes, illustrating the limitation of plan-based methods: because plan-based evaluation cannot anticipate that the agent will get to react to the cue observation, it undervalues information-gathering and prefers the safe option in roughly a third of cases. The native epistemic drive of EFE still sends the agent to the cue in the remaining episodes, and when it does, receding-horizon replanning allows it to exploit the observation, hence the similar cue visit rate (66%) and success rate (61%).

These results validate two claims. First, epistemic priors are necessary for reliable information-seeking: without them ($F_{\text{marginal}}$, $F_{\text{planning}}$), the agent cannot anticipate the value of visiting the cue. Second, policy-based methods that maintain a joint posterior over states and actions ($F_{\text{active}}$, Sophisticated Inference) outperform plan-based methods (Standard EFE planning) in stochastic environments, because they can anticipate their own future reactivity, making information-gathering consistently valuable and leading to reliable cue visits.

### 5.5.3 MiniGrid DoorKey

Table 3 summarizes the results over 100 episodes.

Both $F_{\text{active}}$ and $F_{\text{planning}}$ achieve 89% success, while $F_{\text{marginal}}$ succeeds in only 4% of episodes, confirming that the entropy correction accounting for sequential decision structure is necessary for effective multi-step planning in this environment. Without it, the agent cannot coordinate the sequence of finding the key, unlocking the door, and reaching the goal.

The equal success rates of $F_{\text{active}}$ and $F_{\text{planning}}$ are consistent with the nature of the task. Unlike the T-maze and Reactivity Maze, where a latent context parameter $\theta$ must be discovered through epistemic exploration,

the DoorKey-8x8 challenge is primarily spatial navigation under partial observability. The key and door locations are revealed through the agent's field of view during navigation rather than through a dedicated cue. In this setting, the entropy correction of $F_{\text{planning}}$ is sufficient for effective planning, and the epistemic priors of $F_{\text{active}}$ neither help nor hinder. The central contribution of this experiment is the demonstration that the variational formulation with temporal factorization operates successfully in an environment where tabular active inference methods cannot.

## 6 Discussion

### 6.1 From Plans to Policies

The Expected Free Energy $G(\boldsymbol{u})$ is traditionally defined as a cost function for evaluating plans, where a plan $\boldsymbol{u} = (u_1, \dots, u_T)$ is a fixed sequence of actions. Traditional EFE-based methods compute $G(\boldsymbol{u})$ for each candidate plan independently. This plan-based approach does not maintain a joint distribution over states and actions, and therefore cannot represent contingent action selection.

When the transition dynamics $p(x_t|x_{t-1}, u_t)$ are stochastic (i.e., not a delta function), plan-based evaluation becomes fundamentally limited (Lázaro-Gredilla et al., 2024, Appendix I). Suppose an agent faces a goal that can be reached via two different routes, both requiring an initial action $u_1$. After taking $u_1$, a stochastic state transition may lead to one of two intermediate states. From each intermediate state, a different second action $u_2^{(1)}$ or $u_2^{(2)}$ leads efficiently to the goal. The optimal course of action would be to take $u_1$ first, then adapt the second action to whichever state is realized.

However, plan-based EFE evaluation considers each complete action sequence independently. The plan $(u_1, u_2^{(1)})$ may have high probability of failure if the state transition leads to the other branch. Similarly, $(u_1, u_2^{(2)})$ fails if the transition goes the other way. Consequently, both plans involving $u_1$ receive high (unfavorable) EFE costs. Plan-based methods cannot represent the strategy of taking $u_1$ and then adapting, because they evaluate fixed action sequences that commit to specific future actions regardless of what state is reached. The value of the initial action $u_1$ becomes underestimated because all plans containing it appear poor when evaluated independently.

A policy-based approach represents action selection through state-conditioned distributions $\pi(u_t|x_{t-1})$, which specify what action to take for each possible state. This enables contingent action selection: different actions can be taken depending on which state is realized. Our key insight is that by performing inference over the joint posterior $q(\boldsymbol{y}, \boldsymbol{x}, \boldsymbol{u}, \theta)$, we implicitly optimize these policy distributions through marginalization. When we marginalize the joint posterior to obtain:

$$q(u_t|x_{t-1}) = \frac{q(u_t, x_{t-1})}{q(x_{t-1})}, \quad \text{where} \quad q(u_t, x_{t-1}) = \sum_{\boldsymbol{y}, \boldsymbol{x}_{\neq t-1}, \boldsymbol{u}_{\neq t}, \theta} q(\boldsymbol{y}, \boldsymbol{x}, \boldsymbol{u}, \theta), \tag{19}$$

the resulting distribution accounts for all possible future state transitions and the agent's planned responses to each. Crucially, when computing the value of the initial action $u_1$, marginalization accounts for all possible future trajectories. Even though every specific plan starting with $u_1$ may individually have low probability of success, the marginal distribution $q(u_1|x_0)$ can correctly assign high probability to $u_1$ precisely because it factors in the joint value of the initial action paired with all possible adaptive responses to the stochastic outcomes that follow. The policy representation thus captures future adaptability within a single planning step before any actions are executed.

This time-dependent policy emerges naturally from the factorization structure of the variational posterior without requiring explicit recursive modeling or procedural algorithms. The within-horizon adaptability is implicit in the joint inference: marginalizing over all future trajectories automatically accounts for all possible outcomes and the agent's anticipated responses.

The Sophisticated Inference framework (Friston et al., 2021) achieves a similar effect through recursive modeling of belief evolution. By explicitly representing how beliefs update at future timesteps, Sophisticated Inference implicitly represents state-contingent action selection. Our variational formulation provides an

alternative route to the same policy-based objective: through marginalization of a joint posterior rather than recursive belief modeling.

Our T-maze experiments in Section 5.5.1 test the epistemic value contribution in a deterministic setting. The Reactivity Maze experiments in Section 5.5.2 validate the plan-versus-policy distinction empirically: in the stochastic Reactivity Maze, policy-based methods ($F_{\text{active}}$, Sophisticated Inference) achieve near-perfect performance, while the plan-based Standard EFE method visits the cue in only 66% of episodes, preferring the safe option otherwise, because it cannot anticipate being able to react to the cue observation. The MiniGrid experiment in Section 5.5.3 extends this validation to a standard reinforcement learning benchmark where the tabular methods used by Standard EFE planning and Sophisticated Inference cannot operate.

## 6.2 Interpretation of the Epistemic Priors

The epistemic priors in Proposition 1 are not chosen to encode intuitive a priori preferences, but rather are constructed so that the VFE decomposes into expected plan costs (the EFE) plus complexity. Each prior corresponds to a specific term in the EFE decomposition:

- The prior $\tilde{p}(\boldsymbol{u}) \propto \exp(\mathbb{H}\left[q(\boldsymbol{x}|\boldsymbol{u})\right])$ biases action selection toward plans that maintain high entropy over future states, reflecting a preference for maintaining flexibility. When $\mathbb{H}\left[q(\boldsymbol{x}|\boldsymbol{u})\right]$ is high, the agent keeps multiple future states open rather than committing to a narrow set of outcomes, which can be interpreted as a preference for adaptability in the face of uncertain dynamics.

- The prior $\tilde{p}(\boldsymbol{x}) \propto \exp(\mathbb{E}_{q(\theta|\boldsymbol{x})}\left[-\mathbb{H}\left[q(\boldsymbol{y}|\boldsymbol{x}, \theta)\right]\right])$ introduces the ambiguity term, biasing inference toward state trajectories that predict a low-entropy distribution over future observations, under the current uncertainty over $\theta$, which can be interpreted as a preference for states that lead to well-predicted observations.

- The prior $\tilde{p}(\boldsymbol{y}, \boldsymbol{x}) \propto \exp(\mathbb{D}_{\text{KL}}\left[q(\theta|\boldsymbol{y}, \boldsymbol{x})\|q(\theta|\boldsymbol{x})\right])$ introduces the novelty term, favoring observation-state pairs that maximize information gain about parameters.

A notable feature of these priors is that they depend on the variational posterior $q$. This $q$-dependence is a consequence of encoding epistemic value, which is inherently defined in terms of how observations would update beliefs. In practice, the $q$-dependence means that the optimization defines a fixed-point problem: the goal is to find $q^*$ that minimizes the VFE computed with priors that themselves depend on $q^*$. Our convergence analysis in Appendix B shows that standard gradient-based optimization converges reliably across all tested environments.

## 6.3 Scalability

The T-maze experiment in Section 5 performs inference over a joint posterior $q(\boldsymbol{y}, \boldsymbol{x}, \boldsymbol{u}, \theta) = q(\boldsymbol{y}, \theta|\boldsymbol{x})q(\boldsymbol{x}|\boldsymbol{u})q(\boldsymbol{u})$ using a tabular representation. In this case, the dimensionality grows exponentially with the planning horizon $T$. For the T-maze with $|\mathcal{Y}| = 2$ observation values, $|\mathcal{X}| = 5$ states, $|\mathcal{U}| = 4$ actions, $|\Theta| = 2$ context values, and $T = 4$, the joint posterior requires $(|\mathcal{Y}|^T \times |\Theta| \times |\mathcal{X}|^T) + (|\mathcal{X}|^T \times |\mathcal{U}|^T) + |\mathcal{U}|^T = 180,256$ parameters.

The Reactivity Maze and MiniGrid experiments demonstrate how temporal factorization addresses scalability. The temporal factorization in (18) decomposes the joint posterior into per-timestep conditional factors, reducing the parameter count from exponential to linear in the horizon $T$. This factorization is a concrete instance of exploiting the generative model's structure: the Markov property of the transition model directly suggests the per-timestep factors $q(x_t|x_{t-1}, u_t, \theta)$ and $q(u_t|x_{t-1})$. The MiniGrid DoorKey-8x8 experiment (Section 5.5.3) demonstrates this concretely: the environment's state and observation spaces are too large for tabular methods such as Standard EFE planning and Sophisticated Inference, yet the variational formulation with temporal factorization achieves 89% success rate.

More generally, recasting EFE-based planning as variational inference makes the problem amenable to the same approximation techniques that have enabled variational inference to scale in other domains. Mes-

sage passing on factor graphs, amortized inference, and structured mean-field approximations offer natural directions for future work.

## 7    Conclusion

We have presented a variational formulation of Expected Free Energy-based planning. Our main result (Proposition 1) shows that minimizing a Variational Free Energy functional over a generative model augmented with specific epistemic priors yields a natural decomposition into expected plan costs and complexity. This establishes that EFE-based planning can be understood as variational optimization, achieving theoretical alignment with the Free Energy Principle.

The epistemic priors introduced in our formulation encode preferences for ambiguity-minimizing and novelty-seeking behavior. Empirical validation on a deterministic T-maze, a stochastic Reactivity Maze, and the MiniGrid DoorKey-8x8 environment confirms that these priors are sufficient to induce the information-gathering behavior characteristic of active inference agents. On the Reactivity Maze, $F_{\text{active}}$ achieves perfect performance, outperforming both plan-based methods and Sophisticated Inference, while policy-based inference through joint posterior marginalization provides robustness to stochastic transitions that plan-based evaluation lacks. The MiniGrid experiment further demonstrates that the variational formulation scales to environments where existing tabular active inference methods cannot operate.

A key advantage of the variational formulation is that the full joint posterior can be marginalized to obtain state-conditioned action beliefs, effectively yielding a policy rather than a fixed plan. This policy implicitly accounts for the agent's ability to adapt future actions to whatever state actually occurs, providing robustness to stochastic dynamics that plan-based evaluation lacks. Moreover, by recasting EFE minimization as variational optimization, we unlock access to the extensive toolkit of scalable variational methods, including message passing on factor graphs, amortized inference, and stochastic optimization, providing a principled path toward active inference agents that can operate in complex, high-dimensional environments.

### Acknowledgments

This publication is part of the ROBUST project with project number KICH3.LTP.20.006, which is (partly) financed by the Dutch Research Council (NWO), GN Hearing, and the Dutch Ministry of Economic Affairs and Climate Policy (EZK) under the program LTP KIC 2020-2023. This project is also partly financed by Holland High Tech with PPS funding for the AUTO-AR project RVO TKI2112P09.

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

# A Proof of Proposition 1

*Proof of Proposition 1.*

$$F[q] = \mathbb{E}_{q(\boldsymbol{y},\boldsymbol{x},\boldsymbol{u},\theta)} \left[ \log \frac{q(\boldsymbol{y},\boldsymbol{x},\boldsymbol{u},\theta)}{p(\boldsymbol{y},\boldsymbol{x},\boldsymbol{u},\theta)\hat{p}(\boldsymbol{x})\tilde{p}(\boldsymbol{u})\tilde{p}(\boldsymbol{x})\tilde{p}(\boldsymbol{y},\boldsymbol{x})} \right] \tag{20a}$$

$$= \mathbb{E}_{q(\boldsymbol{u})} \left[ \log \frac{q(\boldsymbol{u})}{p(\boldsymbol{u})} + \underbrace{\mathbb{E}_{q(\boldsymbol{y},\boldsymbol{x},\theta|\boldsymbol{u})} \left[ \log \frac{q(\boldsymbol{y},\boldsymbol{x},\theta|\boldsymbol{u})}{p(\boldsymbol{y},\boldsymbol{x},\theta|\boldsymbol{u})\hat{p}(\boldsymbol{x})\tilde{p}(\boldsymbol{u})\tilde{p}(\boldsymbol{x})\tilde{p}(\boldsymbol{y},\boldsymbol{x})} \right]}_{C(\boldsymbol{u})} \right] \tag{20b}$$

$$= \mathbb{E}_{q(\boldsymbol{u})} \left[ \log \frac{q(\boldsymbol{u})}{p(\boldsymbol{u})} + \underbrace{G(\boldsymbol{u}) + \mathbb{E}_{q(\boldsymbol{y},\boldsymbol{x},\theta|\boldsymbol{u})} \left[ \log \frac{q(\boldsymbol{y},\boldsymbol{x},\theta|\boldsymbol{u})}{p(\boldsymbol{y},\boldsymbol{x},\theta|\boldsymbol{u})} \right] + \text{constant}}_{C(\boldsymbol{u}) \text{ if (6) holds}} \right] \tag{20c}$$

$$= \mathbb{E}_{q(\boldsymbol{u})} \left[ G(\boldsymbol{u}) \right] + \mathbb{E}_{q(\boldsymbol{y},\boldsymbol{x},\boldsymbol{u},\theta)} \left[ \log \frac{q(\boldsymbol{y},\boldsymbol{x},\boldsymbol{u},\theta)}{p(\boldsymbol{y},\boldsymbol{x},\boldsymbol{u},\theta)} \right] + \text{constant} \quad \text{if (6) holds} \tag{20d}$$

$\square$

In the above derivation, we still need to prove the transition for $C(\boldsymbol{u})$ from (20b) to (20c), which we address next. In the following, all expectations are with respect to $q(\boldsymbol{y},\boldsymbol{x},\theta|\boldsymbol{u})$ unless otherwise indicated.

**Lemma 1** (Proof of equivalence $C(\boldsymbol{u})$ in (20b) and (20c))**.**

*Proof.*

$$C(\boldsymbol{u}) = \mathbb{E} \left[ \log \frac{\overbrace{q(\boldsymbol{y},\boldsymbol{x},\theta|\boldsymbol{u})}^{\text{posterior}}}{\underbrace{p(\boldsymbol{y},\boldsymbol{x},\theta|\boldsymbol{u})}_{\text{predictive}} \underbrace{\hat{p}(\boldsymbol{x})}_{\text{utility}} \underbrace{\tilde{p}(\boldsymbol{u})\tilde{p}(\boldsymbol{x})\tilde{p}(\boldsymbol{y},\boldsymbol{x})}_{\text{epistemic priors}}} \right] \tag{21a}$$

$$= \mathbb{E} \left[ \log \left( \underbrace{\frac{q(\boldsymbol{x}|\boldsymbol{u})}{\hat{p}(\boldsymbol{x})}}_{\text{risk}} \cdot \underbrace{\frac{1}{q(\boldsymbol{y}|\boldsymbol{x},\theta)}}_{\text{ambiguity}} \cdot \underbrace{\frac{q(\theta|\boldsymbol{x})}{q(\theta|\boldsymbol{y},\boldsymbol{x})}}_{-\text{novelty}} \right) \right] + \tag{21b}$$

$$\underbrace{\phantom{= \mathbb{E} \left[ \log \left( \frac{q(\boldsymbol{x}|\boldsymbol{u})}{\hat{p}(\boldsymbol{x})} \cdot \frac{1}{q(\boldsymbol{y}|\boldsymbol{x},\theta)} \cdot \frac{q(\theta|\boldsymbol{x})}{q(\theta|\boldsymbol{y},\boldsymbol{x})} \right) \right]}}_{G(\boldsymbol{u})=\text{Expected Free Energy}}$$

$$+ \mathbb{E} \left[ \log \left( \underbrace{\frac{\hat{p}(\boldsymbol{x})q(\boldsymbol{y}|\boldsymbol{x},\theta)q(\theta|\boldsymbol{y},\boldsymbol{x})}{q(\boldsymbol{x}|\boldsymbol{u})q(\theta|\boldsymbol{x})}}_{\text{inverse factors from } G(\boldsymbol{u})} \cdot \underbrace{\frac{q(\boldsymbol{y},\boldsymbol{x},\theta|\boldsymbol{u})}{p(\boldsymbol{y},\boldsymbol{x},\theta|\boldsymbol{u})\hat{p}(\boldsymbol{x})\tilde{p}(\boldsymbol{u})\tilde{p}(\boldsymbol{x})\tilde{p}(\boldsymbol{y},\boldsymbol{x})}}_{\text{factors from (21a)}} \right) \right]$$

$$= G(\boldsymbol{u}) + \underbrace{\mathbb{E} \left[ \log \frac{q(\boldsymbol{y},\boldsymbol{x},\theta|\boldsymbol{u})}{p(\boldsymbol{y},\boldsymbol{x},\theta|\boldsymbol{u})} \right]}_{=B(\boldsymbol{u})} + \underbrace{\mathbb{E} \left[ \log \frac{q(\boldsymbol{y}|\boldsymbol{x},\theta)q(\theta|\boldsymbol{y},\boldsymbol{x})}{q(\boldsymbol{x}|\boldsymbol{u})q(\theta|\boldsymbol{x})\tilde{p}(\boldsymbol{u})\tilde{p}(\boldsymbol{x})\tilde{p}(\boldsymbol{y},\boldsymbol{x})} \right]}_{\text{choose epistemic priors to let this be constant}} \tag{21c}$$

$$= G(\boldsymbol{u}) + B(\boldsymbol{u}) + \tag{21d}$$

$$+ \mathbb{E} \left[ \log \frac{1}{q(\boldsymbol{x}|\boldsymbol{u})\tilde{p}(\boldsymbol{u})} \right] + \mathbb{E} \left[ \log \frac{q(\boldsymbol{y}|\boldsymbol{x},\theta)}{\tilde{p}(\boldsymbol{x})} \right] + \mathbb{E} \left[ \log \frac{q(\theta|\boldsymbol{y},\boldsymbol{x})}{q(\theta|\boldsymbol{x})\tilde{p}(\boldsymbol{y},\boldsymbol{x})} \right]$$

$$= G(\boldsymbol{u}) + B(\boldsymbol{u}) + \quad \text{(expand by using factorization (5))} \tag{21e}$$

$$+ \sum_{\boldsymbol{x}} q(\boldsymbol{x}|\boldsymbol{u}) \Big( -\log q(\boldsymbol{x}|\boldsymbol{u}) - \log \tilde{p}(\boldsymbol{u}) \Big)$$

$$+ \sum_{\boldsymbol{x}} q(\boldsymbol{x}|\boldsymbol{u}) \sum_{\boldsymbol{y},\theta} q(\boldsymbol{y},\theta|\boldsymbol{x}) \Big( \log q(\boldsymbol{y}|\boldsymbol{x},\theta) - \log \tilde{p}(\boldsymbol{x}) \Big)$$

$$+ \sum_{\boldsymbol{x}} q(\boldsymbol{x}|\boldsymbol{u}) \sum_{\boldsymbol{y}} q(\boldsymbol{y}|\boldsymbol{x}) \sum_{\theta} q(\theta|\boldsymbol{y},\boldsymbol{x}) \Big( \log \frac{q(\theta|\boldsymbol{y},\boldsymbol{x})}{q(\theta|\boldsymbol{x})} - \log \tilde{p}(\boldsymbol{y},\boldsymbol{x}) \Big)$$

$$= G(\boldsymbol{u}) + B(\boldsymbol{u}) + \quad \text{(move epistemic priors out of sums)} \tag{21f}$$

$$\underbrace{- \log \tilde{p}(\boldsymbol{u}) - \sum_{\boldsymbol{x}} q(\boldsymbol{x}|\boldsymbol{u}) \log q(\boldsymbol{x}|\boldsymbol{u})}_{=\mathbb{H}[q(\boldsymbol{x}|\boldsymbol{u})]}$$

$$+ \sum_{\boldsymbol{x}} q(\boldsymbol{x}|\boldsymbol{u}) \Big( -\log \tilde{p}(\boldsymbol{x}) + \underbrace{\sum_{\boldsymbol{y},\theta} q(\boldsymbol{y},\theta|\boldsymbol{x}) \log q(\boldsymbol{y}|\boldsymbol{x},\theta)}_{=\mathbb{E}_{q(\theta|\boldsymbol{x})}[-\mathbb{H}[q(\boldsymbol{y}|\boldsymbol{x},\theta)]]} \Big)$$

$$+ \sum_{\boldsymbol{x}} q(\boldsymbol{x}|\boldsymbol{u}) \sum_{\boldsymbol{y}} q(\boldsymbol{y}|\boldsymbol{x}) \Big( -\log \tilde{p}(\boldsymbol{y},\boldsymbol{x}) + \underbrace{\sum_{\theta} q(\theta|\boldsymbol{y},\boldsymbol{x}) \Big( \log \frac{q(\theta|\boldsymbol{y},\boldsymbol{x})}{q(\theta|\boldsymbol{x})} \Big)}_{=\mathbb{D}_{\mathrm{KL}}[q(\theta|\boldsymbol{y},\boldsymbol{x})\|q(\theta|\boldsymbol{x})]} \Big)$$

$$= G(\boldsymbol{u}) + B(\boldsymbol{u}) + \tag{21g}$$

$$\underbrace{- \log \tilde{p}(\boldsymbol{u}) + \mathbb{H}\left[q(\boldsymbol{x}|\boldsymbol{u})\right]}_{=\text{const if } \tilde{p}(\boldsymbol{u}) \propto \exp(\mathbb{H}[q(\boldsymbol{x}|\boldsymbol{u})])}$$

$$+ \sum_{\boldsymbol{x}} q(\boldsymbol{x}|\boldsymbol{u}) \Big( \underbrace{- \log \tilde{p}(\boldsymbol{x}) - \mathbb{E}_{q(\theta|\boldsymbol{x})}\left[\mathbb{H}\left[q(\boldsymbol{y}|\boldsymbol{x},\theta)\right]\right]}_{=\text{const if } \tilde{p}(\boldsymbol{x}) \propto \exp(-\mathbb{E}_{q(\theta|\boldsymbol{x})}[\mathbb{H}[q(\boldsymbol{y}|\boldsymbol{x},\theta)]])} \Big)$$

$$+ \sum_{\boldsymbol{x}} q(\boldsymbol{x}|\boldsymbol{u}) \sum_{\boldsymbol{y}} q(\boldsymbol{y}|\boldsymbol{x}) \Big( \underbrace{- \log \tilde{p}(\boldsymbol{y},\boldsymbol{x}) + \mathbb{D}_{\mathrm{KL}}\left[q(\theta|\boldsymbol{y},\boldsymbol{x})\|q(\theta|\boldsymbol{x})\right]}_{=\text{const if } \tilde{p}(\boldsymbol{y},\boldsymbol{x}) \propto \exp(\mathbb{D}_{\mathrm{KL}}[q(\theta|\boldsymbol{y},\boldsymbol{x})\|q(\theta|\boldsymbol{x})])} \Big)$$

$$= G(\boldsymbol{u}) + \mathbb{E}_{q(\boldsymbol{y},\boldsymbol{x},\theta|\boldsymbol{u})} \left[ \log \frac{q(\boldsymbol{y},\boldsymbol{x},\theta|\boldsymbol{u})}{p(\boldsymbol{y},\boldsymbol{x},\theta|\boldsymbol{u})} \right] + \text{constant} \quad \text{if (6) holds.} \tag{21h}$$

$$\square$$

The term in (21g) works out as follows:

$$- \log \tilde{p}(\boldsymbol{u}) + \mathbb{H}\left[q(\boldsymbol{x}|\boldsymbol{u})\right] = \begin{cases} 0 & \text{if } \tilde{p}(\boldsymbol{u}) = \exp(\mathbb{H}\left[q(\boldsymbol{x}|\boldsymbol{u})\right]) \\ \text{const} & \text{if } \tilde{p}(\boldsymbol{u}) = \sigma(\mathbb{H}\left[q(\boldsymbol{x}|\boldsymbol{u})\right]) \end{cases} \tag{22a}$$

where $\text{const} = \log\left(\sum_{\boldsymbol{u}'} \exp(\mathbb{H}\left[q(\boldsymbol{x}|\boldsymbol{u}')\right])\right)$ and

$$\sigma(\mathbb{H}\left[q(\boldsymbol{x}|\boldsymbol{u})\right]) \triangleq \frac{\exp(\mathbb{H}\left[q(\boldsymbol{x}|\boldsymbol{u})\right])}{\sum_{\boldsymbol{u}'} \exp(\mathbb{H}\left[q(\boldsymbol{x}|\boldsymbol{u}')\right])} \tag{23}$$

is the (normalized) softmax function, and similar derivations apply to the other epistemic priors in (6).

In the context of variational inference with Variational Free Energy $F[q]$ as in (4), the normalization of $\tilde{p}(\boldsymbol{u})$ is inconsequential, as the additive constant does not affect the location of the minimum of $F[q]$. As long as $\tilde{p}(\boldsymbol{u}) \propto \exp(\mathbb{H}\left[q(\boldsymbol{x}|\boldsymbol{u})\right])$, the results of VFE minimization will be the same.

# B    Convergence Diagnostics

This appendix investigates the practical optimization properties of the Variational Free Energy objectives introduced in Section 5. We analyze convergence behavior, sensitivity to hyperparameters, and the emergence of epistemic behavior during optimization. Results are presented for both the T-maze and Reactivity Maze environments.

## B.1    Convergence curves

Figure 3 shows the Variational Free Energy as a function of optimization steps for the T-maze ($T = 4$, left) and Reactivity Maze ($T = 3$, right), each averaged over 20 episodes. On the T-maze, all three objectives converge monotonically within approximately 500 iterations, with low variance across episodes (shaded $\pm 1$ standard deviation bands), indicating a well-behaved optimization landscape. The ordering $F_{\mathrm{marginal}} < F_{\mathrm{planning}} < F_{\mathrm{active}}$ reflects the non-negative epistemic prior contributions. As shown by Lázaro-Gredilla et al. (2024, Equation 8), the entropy correction terms in $F_{\mathrm{planning}}$ are always non-negative, so $F_{\mathrm{planning}}$ provides an upper bound on $F_{\mathrm{marginal}}$. The values are not directly comparable across objectives, as each minimizes a different functional.

On the Reactivity Maze, the same ordering holds, and all three objectives converge rapidly within the first 200 iterations. Unlike the T-maze, $F_{\mathrm{active}}$ does not exhibit a prolonged secondary phase, consistent with the observation that 100 optimization steps already suffice for 100% success (Figure 6).

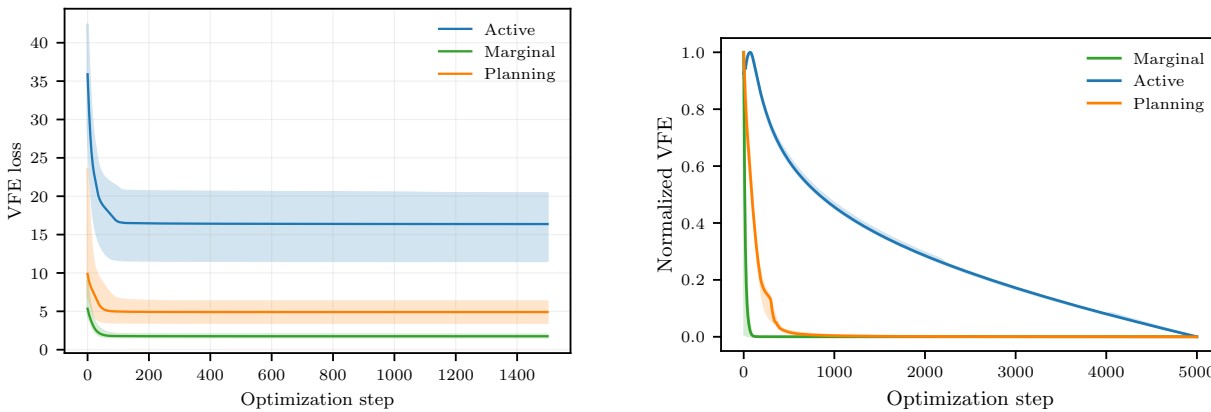

Figure 3: Variational Free Energy during optimization for each objective. T-maze (left), averaged over 20 episodes with $\pm 1$ standard deviation bands. Reactivity Maze (right), normalized per objective.

## B.2    Two-phase optimization

Figure 4 compares the T-maze convergence curves for the first planning step under two conditions: unknown parameters ($\theta$ unknown, left) and known parameters ($\theta$ known, right). When $\theta$ is unknown, the $F_{\mathrm{active}}$ curve exhibits a notable two-phase structure. In the first phase (steps 0–100), all objectives descend rapidly as the model fits the basic state and observation structure. In the second phase (steps 1000–1200), $F_{\mathrm{active}}$ undergoes a secondary, more gradual descent that is absent in $F_{\mathrm{marginal}}$ and $F_{\mathrm{planning}}$. This secondary drop corresponds to the epistemic priors gradually steering the policy toward information-seeking actions, as confirmed by the policy stability analysis in Figure 5.

When $\theta$ is known, this second phase largely vanishes: with no parameter uncertainty to resolve, the novelty-seeking prior contributes less, and the exploitation structure alone determines convergence. This two-phase structure has practical implications for the required optimization budget, as discussed next.

### B.3 Optimization budget

### B.3.1 T-maze

Although the VFE loss appears converged by approximately 500 iterations (Figure 3), task performance depends on a sufficient optimization budget. Figure 5 (left) shows the success rate as a function of optimization steps. $F_{\text{active}}$ requires approximately 1500 steps to achieve 100% success, while $F_{\text{marginal}}$ reaches 75% at around 1000 steps and $F_{\text{planning}}$ never exceeds 25%.

The policy stability analysis (Figure 5, right) reveals why: the probability of the cue-visiting action (South) for $F_{\text{active}}$ gradually increases from approximately 0.05 at 10 steps to 0.41 at 1500 steps, stabilizing thereafter. This confirms that the VFE loss has a long, shallow tail that, while small in absolute terms, is behaviorally decisive. The exploitation structure is learned quickly, but the exploration signal from the epistemic priors takes longer to dominate. For $F_{\text{marginal}}$ and $F_{\text{planning}}$, $P(\text{South})$ remains near zero regardless of the optimization budget.

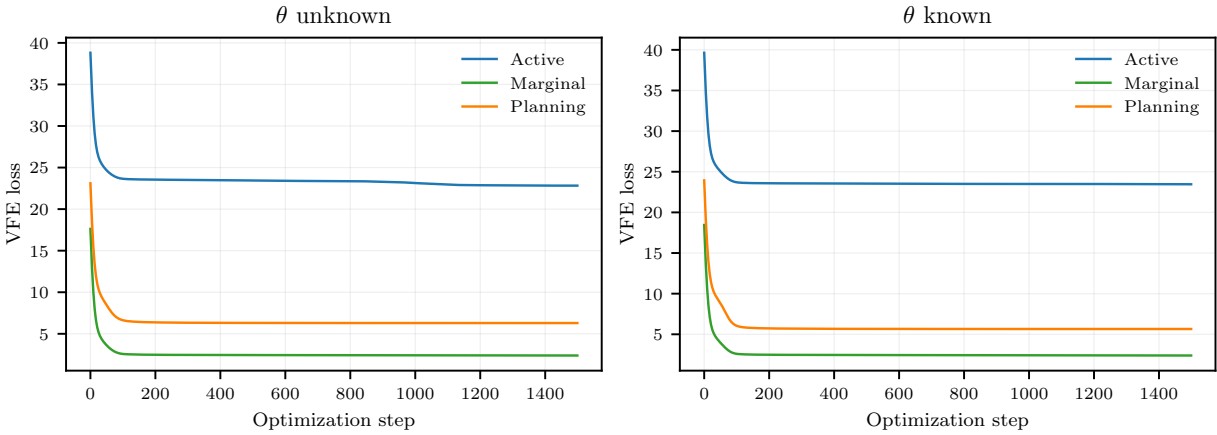

Figure 4: T-maze: convergence curves for the first planning step with unknown parameters (left) and known parameters (right). Under parameter uncertainty, $F_{\text{active}}$ exhibits a secondary descent (steps 1000–1200) corresponding to the emergence of epistemic behavior. This second phase is absent when $\theta$ is known.

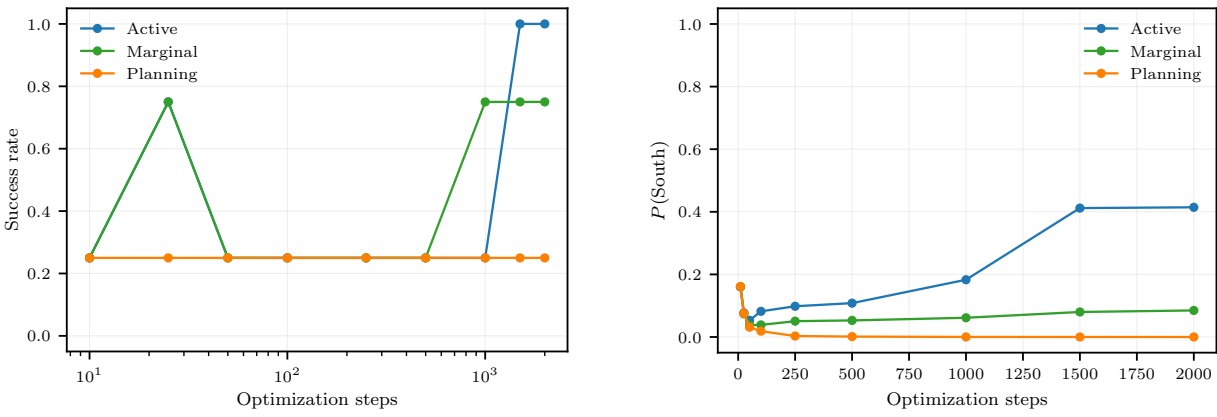

Figure 5: T-maze: success rate as a function of optimization steps (left) and probability of the cue-visiting action at the first time step (right). For $F_{\text{active}}$, the information-seeking policy emerges gradually and stabilizes at approximately 1500 steps, consistent with the two-phase convergence structure observed in Figure 4.

### B.3.2 Reactivity Maze

Figure 6 (left) shows the optimization budget analysis for the Reactivity Maze. $F_{\text{active}}$ achieves 100% success with as few as 100 optimization steps, substantially fewer than the T-maze requires ($\sim$1500). $F_{\text{marginal}}$ remains at 0% success regardless of the budget, and $F_{\text{planning}}$ reaches at most 25%. The sharper transition for $F_{\text{active}}$ suggests that the Reactivity Maze presents a more clear-cut distinction between objectives: information seeking is both necessary and sufficient, and the optimization landscape resolves this quickly.

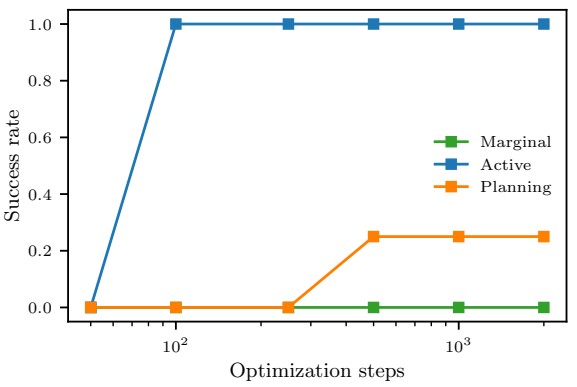 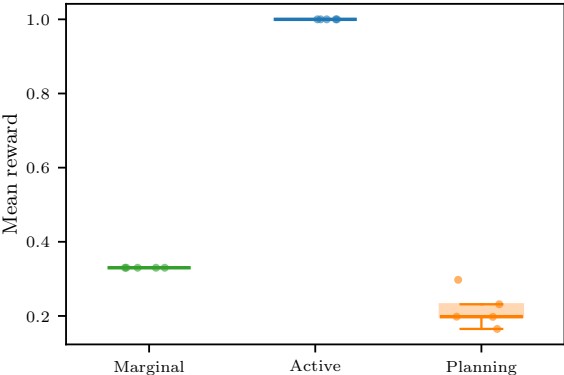

Figure 6: Reactivity Maze: success rate as a function of optimization steps (left) and mean reward across 5 random seeds (right). $F_{\text{active}}$ reaches 100% success at only 100 steps and achieves perfect consistency $(1.0 \pm 0.0)$ across seeds.

## B.4 Learning rate sensitivity and robustness

### B.4.1 T-maze

Figure 7 (left) shows the T-maze success rate as a function of learning rate (1500 optimization steps, 20 episodes). $F_{\text{active}}$ achieves 100% success at a learning rate of 0.05. Smaller learning rates (0.001–0.01) prevent the slow epistemic phase from completing within the optimization budget, while a larger learning rate (0.1) causes the optimizer to overshoot, reducing performance to 75%. $F_{\text{marginal}}$ is more robust, achieving 75% success at both low and high learning rates. $F_{\text{planning}}$ remains at 25% regardless of learning rate, as it lacks the goal-directed priors needed to solve the task.

Figure 7 (right) reports mean reward across 5 random seeds (20 episodes each). $F_{\text{active}}$ achieves a mean reward of $1.0 \pm 0.0$: once the optimization budget is sufficient, behavior is perfectly consistent across seeds. $F_{\text{marginal}}$ averages $0.14 \pm 0.22$ and $F_{\text{planning}}$ averages $-0.14 \pm 0.22$, both with substantial variance reflecting their reliance on chance rather than directed information seeking.

### B.4.2 Reactivity Maze

In contrast, Figure 8 shows that $F_{\text{active}}$ on the Reactivity Maze achieves 100% success across all tested learning rates (0.005–0.1). This robustness indicates that the T-maze sensitivity is environment-specific rather than a fundamental limitation of the approach. $F_{\text{marginal}}$ remains at 0% success and $F_{\text{planning}}$ reaches at most 40% across all learning rates. The Reactivity Maze seed variance is shown in Figure 6 (right): $F_{\text{active}}$ achieves $1.0 \pm 0.0$, $F_{\text{marginal}}$ is consistent at the random baseline ($0.33 \pm 0.0$), and $F_{\text{planning}}$ averages $0.22 \pm 0.05$.

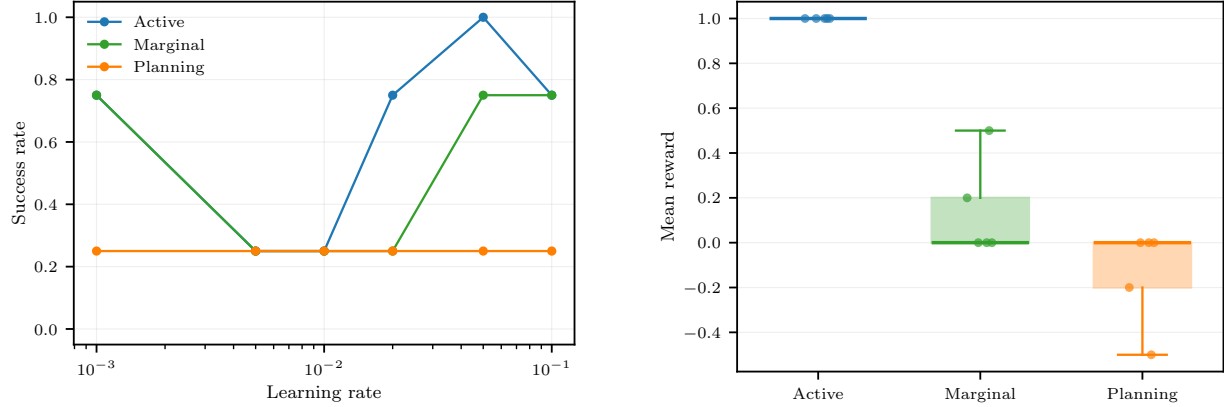

Figure 7: T-maze: success rate as a function of learning rate (left) and mean reward across 5 random seeds (right). $F_{\text{active}}$ achieves full success only at a learning rate of 0.05, but is perfectly consistent $(1.0 \pm 0.0)$ across seeds once converged.

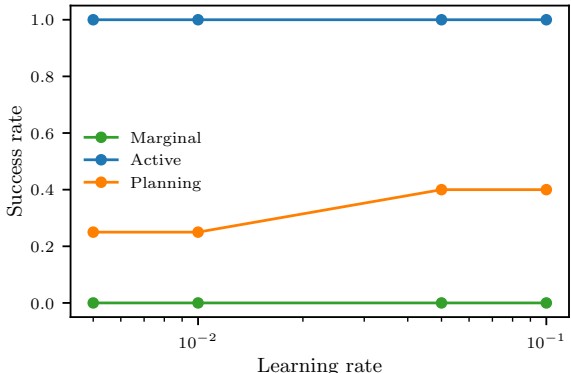

Figure 8: Reactivity Maze: success rate as a function of learning rate. $F_{\text{active}}$ achieves 100% success across all tested learning rates, in contrast to the T-maze sensitivity.

# C  MiniGrid Convergence Analysis

This appendix examines the convergence properties of the Variational Free Energy objectives on the MiniGrid DoorKey environment (Section 5.1.3), which presents a substantially larger state and observation space than the T-maze and Reactivity Maze. We compare two optimizers, Adam (Kingma & Ba, 2015) and Adafactor (Shazeer & Stern, 2018), across different knowledge conditions and planning horizons. All experiments use an $8 \times 8$ grid ($6 \times 6$ internal grid) with $3 \times 3$ field of view, a learning rate of 0.01, and 3000 optimization steps per planning step.

## C.1  Knowledge conditions

Figure 9 shows convergence trajectories for four knowledge scenarios of increasing prior information: *tabula rasa* (no prior knowledge), *known position* (agent knows its location), *known layout* (agent knows the grid structure), and *near goal* (agent starts close to the goal). Each row corresponds to a different objective, with Adam (solid) and Adafactor (dashed) compared within each panel.

$F_{\mathrm{marginal}}$ (top row) converges rapidly and identically for both optimizers across all scenarios, reaching its minimum within approximately 100 steps. $F_{\mathrm{active}}$ (middle row) exhibits qualitatively different behavior depending on the knowledge condition. In the *tabula rasa* and *known layout* scenarios, the VFE *increases* during optimization, in contrast to the monotonic descent observed in the T-maze and Reactivity Maze (Appendix B). This increase reflects the epistemic priors reshaping the policy toward information-seeking actions, which incur higher expected entropy costs. In the *known position* and *near goal* scenarios, the VFE decreases monotonically, indicating that sufficient prior knowledge reduces the contribution of the epistemic terms. Adafactor fails to converge in some active scenarios (notably *known position*). $F_{\mathrm{planning}}$ (bottom row) shows gradual step-wise decreases, with Adam and Adafactor following different paths to similar final values.

The *near goal* scenario produces substantially smaller loss magnitudes across all objectives, consistent with the reduced planning complexity when the agent starts close to its target.

## C.2  Horizon scaling

Figure 10 compares convergence at planning horizons $T = 20$ and $T = 40$ under the tabula rasa condition. Doubling the horizon roughly doubles the loss magnitudes for $F_{\mathrm{active}}$ and $F_{\mathrm{planning}}$, as expected from the additional time steps in the objective. $F_{\mathrm{marginal}}$ scales cleanly, converging at similar rates for both horizons. For $F_{\mathrm{active}}$ at $T = 40$, Adafactor fails to converge within 3000 steps, while Adam still reaches a stable solution. This suggests that Adam is the more reliable optimizer for the active objective, particularly at longer horizons where the epistemic prior contributions are larger.

In summary, all objectives converge within 3000 optimization steps when using Adam, across all tested knowledge conditions and horizons. The non-monotonic behavior of $F_{\mathrm{active}}$ in certain knowledge conditions suggests a qualitatively different optimization landscape than the monotonic descent observed in the smaller environments (Appendix B). Adam is the preferred optimizer for the active objective due to its reliability, while both optimizers perform comparably for $F_{\mathrm{marginal}}$ and $F_{\mathrm{planning}}$.

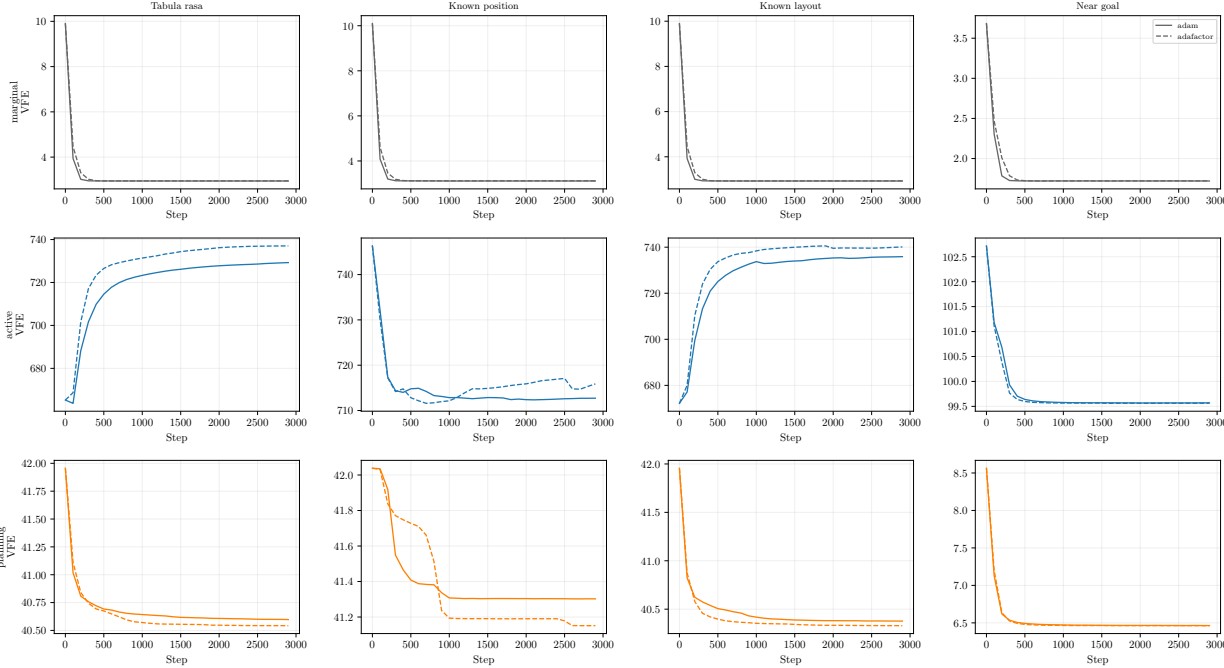

Figure 9: MiniGrid DoorKey: convergence trajectories across knowledge scenarios (columns) and objectives (rows), comparing Adam (solid) and Adafactor (dashed). $F_{\text{active}}$ exhibits non-monotonic behavior that depends on the knowledge condition, while $F_{\text{marginal}}$ converges rapidly and smoothly.

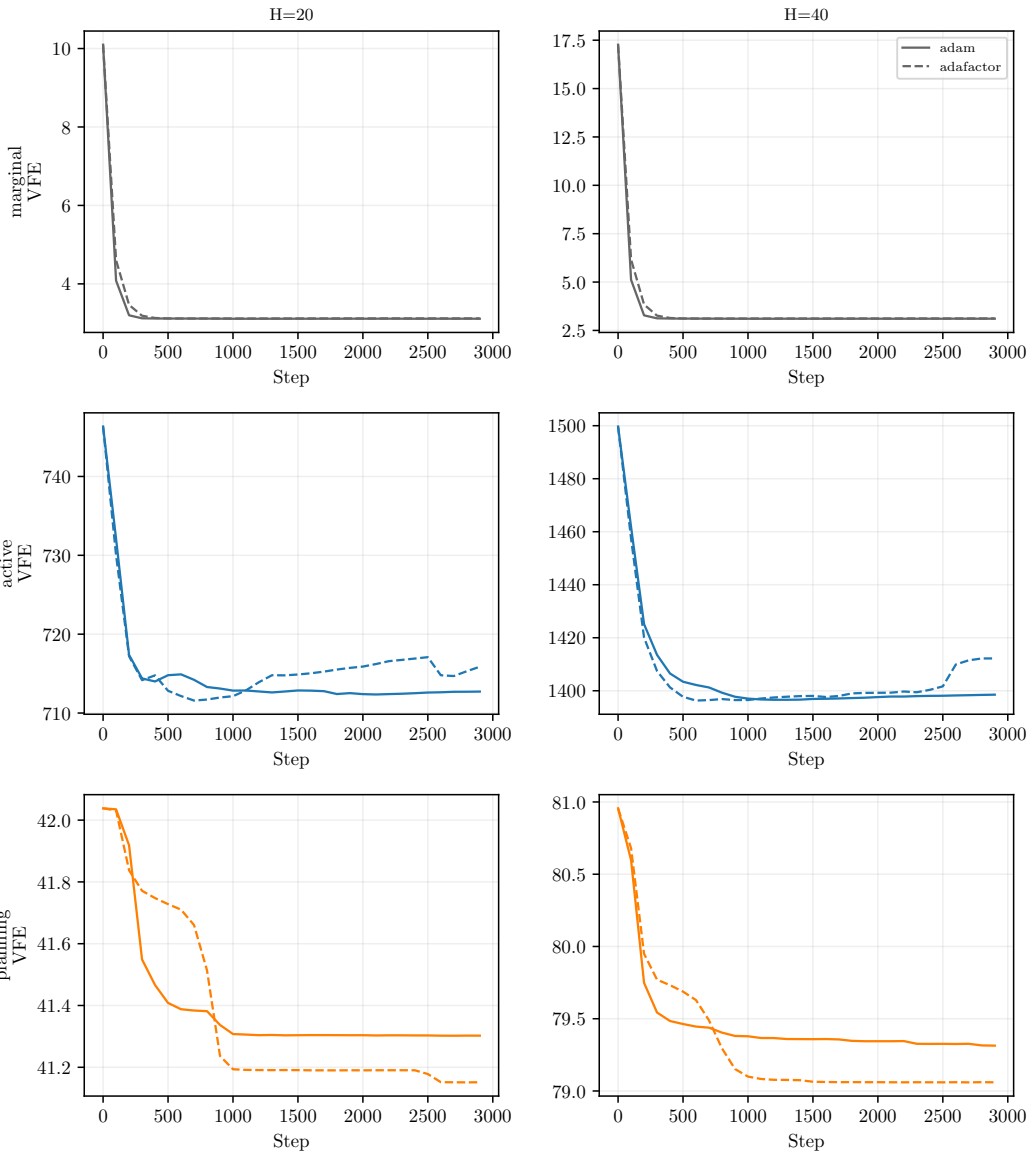

Figure 10: MiniGrid DoorKey: convergence trajectories at planning horizons $T = 20$ (left) and $T = 40$ (right), comparing Adam (solid) and Adafactor (dashed). Adafactor fails to converge for $F_{\text{active}}$ at $T = 40$.

