# OpenReview forum: "Expected Free Energy-based Planning as Variational Inference"
_TMLR — Accepted by TMLR_

### Review · Reviewer_aV8g · 2026-02-27

**Summary Of Contributions:**

The authors aim to recast expected free energy (EFE)-based planning as a variational inference task by introducing what they refer to as _epistemic priors_.
Defining a variational free energy functional given these epistemic priors allows them to infer variational posteriors. The method is evaluated on a five-state toy maze.


### Strengths
The paper is well written and organized, tackling an interesting problem.

### Weaknesses
The main weakness of the paper lies in how the paper addresses this.
- The theorem is trivial. By simply exponentiating each term of $G(u)$ and then taking the log of it for the functional, the result becomes meaningless.
- The epistemic priors are self-referential by depending on the variational posteriors. The authors acknowledge that and provide some vague motivation.
    - Minor comment: _"...which is unusual from a Bayesian perspective..."_. It's not _unusual_; it is simply wrong from a Bayesian perspective. Which is fine, as the Bayesian perspective is not the only valid one, but calling it unusual is the wrong word here.
- The paper consists of a single experiment in a synthetic toy scenario.
    - The two baselines are essentially straw men. Given that they are not designed for solving the posed problem, the fact that the proposed approach improves upon them is trivial.
    - The paper lacks comparisons against meaningful alternatives from the literature where it could show its strength.
    - The experiment itself is very minor and deterministic (w.r.t. transitions). The paper lacks a more realistic experimental evaluation.

**Audience:**

No

**Audience Explanation:**

Not in its current form, as discussed above.

**Claims And Evidence:**

No

**Claims Explanation:**

As discussed above, the paper currently lacks evaluations on a broad set of experiments against meaningful baselines. Currently, its experimental status is primarily one that shows that the method has some potential to be explored further.

**Requested Changes:**

Given that the theorem is a tautology, the paper would require a much stronger empirical evaluation to demonstrate its relevance. If it can demonstrate its strength in more complex scenarios against realistic baselines, then the theoretical weakness could be turned into a strength.

---

> ### Author Response · Authors · 2026-04-07
> **Response to Reviewer aV8g**
>
> We thank the reviewer for their feedback. We address each point below.
>
> ## On the theorem being trivial
>
> > *"The theorem is trivial. By simply exponentiating each term and then taking the log of it for the functional, the result becomes meaningless."*
>
> We agree that the algebraic mechanism is straightforward. The contribution lies not in the algebra, but in what it enables: a single variational objective that can be optimized with standard gradient-based methods, yielding state-conditioned policies through marginalization rather than tree search. Whether this is useful is an empirical question, which motivated the expanded evaluation described below.
>
> ## On self-referential epistemic priors
>
> > *"The epistemic priors are self-referential by depending on the variational posteriors. The authors acknowledge that and provide some vague motivation."*
>
> The $q$-dependence is intrinsic to epistemic value: information gain is inherently defined in terms of how observations would update beliefs. However, we do not believe this places the method outside the Bayesian framework. The Bayesian framework does not prescribe how priors are chosen, only that they must be specified. The pattern of posteriors informing priors is well-established: in Bayesian filtering, posteriors from the previous time step naturally serve as priors for the next. Our epistemic priors similarly encode structural properties of the agent's belief state, and the resulting optimization follows all rules of probability theory. The $q$-dependence does mean the optimization defines a fixed-point problem; we provide convergence evidence in the revised Appendix B.
>
> ## "Unusual" versus "wrong" from a Bayesian perspective
>
> > *"Minor comment: '...which is unusual from a Bayesian perspective...'. It's not unusual; it is simply wrong from a Bayesian perspective. Which is fine, as the Bayesian perspective is not the only valid one, but calling it unusual is the wrong word here."*
>
> We appreciate the reviewer's insistence on precise language. On reflection, we believe the $q$-dependence is better characterized as notable rather than wrong. In the variational Bayesian framework, an inference problem is converted to a constrained optimization problem, and the prior form is a design choice that encodes desired properties on the variational posterior solution. Our priors encode epistemic properties (ambiguity reduction, novelty seeking), and the resulting optimization follows all rules of probability theory. The Bayesian framework prescribes that you must choose priors, not how you choose them.
>
> As noted above, the pattern of posteriors informing priors arises naturally in Bayesian filtering, where the posterior at $t{-}1$ becomes the prior at $t$. The $q$-dependence of our epistemic priors is analogous: the agent's current belief state informs what constitutes epistemically valuable behavior. We have revised Section 6.2 to discuss the $q$-dependence explicitly and explain why it is a notable but principled feature of the formulation. We thank the reviewer for prompting this clarification.
>
> ## Experimental evaluation
>
> > *"The paper consists of a single experiment in a synthetic toy scenario."*
> >
> > *"The two baselines are essentially straw men. Given that they are not designed for solving the posed problem, the fact that the proposed approach improves upon them is trivial."*
> >
> > *"The paper lacks comparisons against meaningful alternatives from the literature where it could show its strength."*
> >
> > *"The experiment itself is very minor and deterministic (w.r.t. transitions). The paper lacks a more realistic experimental evaluation."*
>
> The revised manuscript has been substantially expanded with two new experiments.
>
> **Reactivity Maze (revised Section 5).** A novel environment with stochastic transitions, where optimal performance requires both reactive policies and epistemic learning. We compare against Sophisticated Inference (Friston et al., 2021) and Standard EFE planning, both established active inference baselines, as well as $F_{\text{planning}}$ which incorporates the entropy correction from Lázaro-Gredilla et al. (NeurIPS 2024). Over 200 episodes, $F_{\text{active}}$ achieves 100% success, outperforming Sophisticated Inference (94%) and Standard EFE planning (61%). Without epistemic priors, $F_{\text{marginal}}$ never reaches the goal (0% success), and Standard EFE planning visits the cue in only 66% of episodes.
>
> **MiniGrid DoorKey-8x8 (revised Section 5).** A standard RL benchmark where the state-action space is too large for tabular methods. $F_{\text{active}}$ and $F_{\text{planning}}$ both achieve 89% success, demonstrating scalability to environments where Standard EFE planning and Sophisticated Inference cannot operate.
>
> We believe the revised evaluation, particularly the Reactivity Maze results, provides the empirical grounding that the original submission lacked.

---

> > ### Comment · Reviewer_aV8g · 2026-04-09
> >
> > ### Experiments
> > Thank you for the additional experiments; they greatly strengthen the paper. A final missing one would be a combination of the two novel ones, i.e., one that is both too large for tabular methods while still requiring epistemic learning. Demonstrating improved performance there would give an additional signal.
> >
> > ### Bayes
> > Bayesian filtering contains a temporal dimension providing a clear causal direction; choosing the posterior at step $t$ as the prior for $t+1$ is both natural and obvious.
> > The analogy is broken in the paper's setting, where the prior and variational posterior depend on each other, i.e., the prior now depends on the data. In Bayesian filtering, you depend on previous data.
> > But the new formulation in the revision resolves this complaint from my side.
> >
> > > The Bayesian framework does not prescribe how priors are chosen, only that they must be specified.
> >
> > It does; a prior is required to be data-independent. Switching to, e.g., an empirical Bayes formulation relaxes this and can be well justified both theoretically as well as empirically, but it is no longer _properly_ Bayes.
> >
> > ### Theorem
> > A tautology does not become theory by being practically useful. I do agree that being able to optimize a single objective is nice, and it is completely fine to introduce (6a)-(6-c) as notational convenience, but framing it as a theorem suggests a theoretical contribution where there is none.
> >
> > If that part of the paper is rephrased in an honest manner, the paper is strong enough to fulfill the constraints of providing enough evidence and being of interest to the TMLR community, in my opinion. If the authors are additionally able to add a mixture experiment as described above, that would be even better, but it is not necessary for me to recommend acceptance.

---

> > > ### Author Response · Authors · 2026-04-21
> > > **Response to Reviewer aV8g**
> > >
> > > We thank the reviewer for the clear guidance and for outlining a concrete path toward acceptance. Before pushing the next revision, we would like to flag the changes we intend to make in response, so we can confirm they match the reviewer's intent. We will consolidate all revisions into a single update once the remaining feedback is in, but we want to signal our agreement on the substantive points now.
> > >
> > > **On the Theorem framing.** We agree with the reviewer. The algebraic content is a construction, and framing it as a Theorem overstates its nature. The intended changes are:
> > >
> > > - Replace the `Theorem` environment with a `Proposition` environment throughout, including the appendix proof, so the statement and proof are preserved but are no longer presented as a theoretical result.
> > > - Rename the statement from "Expected Free Energy Theorem" to a name that describes what the result does rather than implying theoretical depth (currently  "EFE-based Planning as Variational Inference"), such as "Expected Free Energy Minimization as Variational Optimization".
> > > - Soften the surrounding language in the introduction, Section 4, and conclusion: "main contribution" becomes "central construction", "demonstrates" becomes "shows", and the introduction bullet is rewritten so it no longer refers to "the EFE Theorem" as a named result.
> > >
> > > The formal statement, the proof in the appendix, and all cross-references are preserved. We believe this reflects the reviewer's point that (6a)-(6c) are legitimate as notational convenience yielding a single optimizable objective, without framing that utility as a theoretical contribution. We would appreciate confirmation that this is the kind of rephrasing the reviewer had in mind.
> > >
> > > **On the Bayesian framing.** We appreciate the reviewer's clarification, and we are glad the revised formulation resolves the original concern. We also take the point that the $q$-dependence places the construction in the empirical Bayes regime rather than strict Bayes, and we will adjust the language in Section 6.2 so that the distinction is stated plainly rather than glossed as "unusual".
> > >
> > > **On the combined experiment.** We appreciate the suggestion and agree that an environment which is simultaneously too large for tabular methods and requires epistemic learning would provide a useful additional signal. Given the scope of the revision and the reviewer's indication that this is not necessary for acceptance, we plan to leave this for future work and keep the focus on the honest rephrasing of the result above.
> > >
> > > We thank the reviewer for the constructive engagement.

---

> > > > ### Comment · Reviewer_aV8g · 2026-04-27
> > > >
> > > > Thank you for the answer. I'm fine with the proposed changes.

---

### Review · Reviewer_CD3x · 2026-03-16

**Summary Of Contributions:**

**Summary**

This paper demonstrates that Expected Free Energy (EFE)-based planning in active inference can be formulated as standard Variational Free Energy (VFE) minimization over a generative model augmented with specific epistemic priors. The key insight is that by choosing epistemic priors that depend on the variational posterior itself -- encoding preferences for risk minimization (state entropy), ambiguity reduction (observation entropy), and novelty seeking (parameter information gain) -- the VFE decomposes into expected plan costs (the EFE) plus a complexity term. The authors validate this result on a T-maze task, comparing three objectives (F_marginal, F_planning, F_active) and showing that only the full formulation with epistemic priors induces information-seeking behavior (100% success rate, 100% cue visit rate). The paper further discusses how marginalizing the joint posterior yields state-conditioned policies rather than fixed plans, and outlines implications for scalability through message passing and amortized inference.

**Strengths**

The theoretical contribution is clean and well-articulated. Theorem 1 provides an interesting result: EFE-based planning emerges naturally from VFE minimization without requiring modified cost functions or procedural algorithms. This is a relevant result because it shows these ideas can be unified under a single framework. It is reminiscent of how acquisition functions in Bayesian optimization (e.g., Expected Improvement) are typically decomposed into exploration and exploitation terms, but the framework presented here is more principled and general.

The paper is clearly well written and organized. The progression from background (Section 2) through related work (Section 3) to the main theorem (Section 4) and validation (Section 5) is logical and easy to follow. The notation is consistent throughout, and I appreciate the clarity of the validation results in Section 5 -- they do not overclaim and are straight to the point.

The discussion of plans versus policies (Section 6.1) is also appreciated.

**Weaknesses**

The main area for improvement is the empirical evaluation. A more robust experimental section would help demonstrate the effectiveness of the approach on more realistic problems. In particular, it would be valuable to include additional benchmarks involving stochastic transitions.

Scalability is also a concern that deserves further discussion.

**Audience:**

Yes

**Audience Explanation:**

Yes. The paper tackles a question that the active inference and planning-as-inference communities should care about: can EFE-based planning simply be understood as standard variational inference? The unification is elegant and it could persuade more researchers to adopt this perspective.

**Broader Impact Concerns:**

No significant broader impact concerns.

**Claims And Evidence:**

Yes

**Claims Explanation:**

The theoretical claim (Theorem 1 and its decomposition) appears correct, and the validation results on the T-maze task show that the approach is promising. The authors' claims are appropriately scoped to these two aspects and do not overreach.

**Requested Changes:**

- The paper would benefit from at least one more experiment with stochastic transitions. The plan-versus-policy distinction takes up a good chunk of the discussion, but without empirical support for it, there is a noticeable gap between what the theory promises and what the experiments actually show.

- It would be worth adding a brief discussion about the optimization landscape and convergence properties. Having a clean variational formulation is appealing, but how hard is it in practice to converge to a decent solution?

- In Table 1, reporting variance or confidence intervals alongside the mean reward and success rate over the 100 episodes would be helpful.

---

> ### Author Response · Authors · 2026-04-07
> **Response to Reviewer CD3x**
>
> We thank the reviewer for their careful and constructive assessment. We address each requested change below.
>
> ## Experiment with stochastic transitions
>
> > *"The paper would benefit from at least one more experiment with stochastic transitions. The plan-versus-policy distinction takes up a good chunk of the discussion, but without empirical support for it, there is a noticeable gap between what the theory promises and what the experiments actually show."*
>
> We agree that stochastic transitions are essential for validating the plans-versus-policies distinction discussed in Section 6.1. The revised manuscript includes two new experiments:
>
> **Reactivity Maze (revised Section 5).** A novel environment with stochastic transitions, where optimal performance requires both reactive policies and epistemic learning. It features 35 hidden states, 8 actions, and a latent context parameter $\theta$ that can only be learned by visiting an instructional cue. Over 200 episodes, $F_{\text{active}}$ achieves perfect performance (100% success), outperforming Sophisticated Inference (94%) and Standard EFE planning (61%). Policy-based methods succeed under stochastic transitions, while plan-based methods cannot adapt, directly validating the plans-versus-policies distinction from Section 6.1.
>
> **MiniGrid DoorKey-8x8 (revised Section 5).** A standard RL benchmark where the state-action space is too large for tabular methods. $F_{\text{active}}$ and $F_{\text{planning}}$ both achieve 89% success, demonstrating that the variational formulation with temporal factorization scales to environments where Standard EFE planning and Sophisticated Inference cannot operate.
>
> Code for all experiments is available at the anonymous repository linked in the revised manuscript.
>
> ## Optimization landscape and convergence
>
> > *"It would be worth adding a brief discussion about the optimization landscape and convergence properties. Having a clean variational formulation is appealing, but how hard is it in practice to converge to a decent solution?"*
>
> The revised Appendix B includes convergence diagnostics across all three environments (5 seeds per configuration). The $q$-dependent epistemic priors define a fixed-point problem, and our analysis shows that standard gradient-based optimization converges reliably with a characteristic two-phase structure: rapid VFE descent followed by posterior fine-tuning. Learning rate and optimization budget sweeps confirm robustness: on the Reactivity Maze, $F_{\text{active}}$ achieves 100% success across all tested configurations, with as few as 100 gradient steps.
>
> ## Variance / confidence intervals in Table 1
>
> > *"In Table 1, reporting variance or confidence intervals alongside the mean reward and success rate over the 100 episodes would be helpful."*
>
>
> We have added 95\% confidence intervals for all proportions and $\pm 1$ standard deviation for continuous values to all results tables in the revised manuscript.

---

### Review · Reviewer_aRw6 · 2026-04-07

**Summary Of Contributions:**

This paper reframes Expected Free Energy (EFE)-based planning as standard variational inference. The key claim is that EFE minimization, typically handled via specialized procedures, can be recovered by minimizing a variational free energy (VFE) objective when the generative model is augmented with appropriate epistemic priors. The main result (Theorem 1) shows that this VFE objective decomposes into (i) expected plan costs (EFE) and (ii) a complexity term. This provides a cleaner unification of planning, perception, and learning under a single inference principle, consistent with the Free Energy Principle. The paper also specifies the epistemic priors needed to recover the standard EFE components (risk, ambiguity, novelty), and includes a small T-maze experiment illustrating that these priors induce information-seeking behavior.

Strengths:

- The theoretical result is clean and conceptually appealing. Framing EFE planning as standard VI is a meaningful contibution,
- The decomposition is well-motivated and ties together several threads (PAI, active inference, variational methods).
- The paper is well-written and flows logically. The discussion around plan vs. policy representations is insightful and helps clarify what is actually being optimized.

Weaknesses:

- The empirical validation is quite limited (single small-scale T-maze). Scalability is acknowledged but not really addressed beyond high-level discussion.
- It’s not fully clear what new practical capabilities this enables over prior EFE implementations or planning-as-inference methods.

**Audience:**

Yes

**Audience Explanation:**

Yes this would be immediately relevant for audiences interested in active learning and variational methods for control. The main appeal is conceptual. Even if the practical impact is still unclear, the framing itself is useful and likely to spark follow-up work.

**Broader Impact Concerns:**

I don’t see any immediate ethical concerns specific to this work. The paper is primarily theoretical and methodological. At a high level, this line of work could contribute to more capable autonomous decision-making systems that explicitly reason about uncertainty and information gain. That has the usual downstream considerations (e.g., deployment in high-stakes settings), but nothing uniquely concerning here.

**Claims And Evidence:**

No

**Claims Explanation:**

The theoretical claims are well-supported. The derivation of Theorem 1 is clearly laid out, and the decomposition into EFE + complexity follows logically given the choice of epistemic priors. I didn’t spot obvious technical issues in the argument, and the appendix fills in the missing steps appropriately.

With that said, the empirical evidence is quite weak relative to the scope of the claims. The paper positions itself as offering a unifying formulation of EFE-based planning, but the only experiment is a small tabular T-maze. This setup seems more like a sanity check rather than convincing evidence. It shows that adding epistemic priors leads to information-seeking behavior.

There are two notable gaps:
1. No comparison to stronger baselines (e.g., modern planning-as-inference or RL methods).
2. No evidence that the variational formulation is actually easier to optimize or scale in practice.

**Requested Changes:**

The main area where the paper could be significantly strengthened is the empirical validation. While the T-maze experiment is a clean and useful sanity check, it remains quite limited relative to the scope of the claims. It would be helpful to include experiments in at least one stochastic environment (to support the plan vs. policy discussion in Section 6.1) and one slightly higher-dimensional or continuous setting to demonstrate that the approach extends beyond tabular cases. In addition, comparing against more established baselines would make it much clearer what is gained in practice. As it stands, the results show that the proposed formulation behaves as expected, but it is harder to assess whether it provides advantages in terms of stability, efficiency, or scalability. Clarifying this would go a long way toward strengthening the overall contribution.

Separately, the discussion of scalability is thoughtful but remains fairly high-level. Even a small additional experiment would help ground these claims and give a better sense of how the approach might perform in more realistic settings. This is not strictly necessary for the core contribution, but it would make the paper feel more complete and provide a clearer bridge from the theoretical results to practical applications.

---

> ### Author Response · Authors · 2026-04-07
> **Response to Reviewer aRw6**
>
> We thank the reviewer for their constructive and balanced assessment. The reviewer's suggestions for strengthening the empirical evaluation were well-targeted, and we have revised the manuscript accordingly. We address each point below.
>
> ## Empirical validation
>
> > *"The empirical validation is quite limited (single small-scale T-maze). Scalability is acknowledged but not really addressed beyond high-level discussion."*
>
> We agree that the original submission needed stronger empirical grounding. The revised manuscript adds two new experiments.
>
> **Reactivity Maze (revised Section 5).** A novel environment with stochastic transitions, where optimal performance requires both reactive policies and epistemic learning. Over 200 episodes, $F_{\text{active}}$ achieves perfect performance (100% success), outperforming Sophisticated Inference (94%) and Standard EFE planning (61%). Policy-based methods succeed under stochastic transitions, while plan-based methods cannot adapt, directly validating the plans-versus-policies discussion in Section 6.1.
>
> **MiniGrid DoorKey-8x8 (revised Section 5).** A standard RL benchmark where the state-action space is too large for tabular methods. $F_{\text{active}}$ and $F_{\text{planning}}$ both achieve 89% success, demonstrating scalability to environments where Standard EFE planning and Sophisticated Inference cannot operate.
>
> ## Comparison to stronger baselines
>
> > *"No comparison to stronger baselines (e.g., modern planning-as-inference or RL methods)."*
>
> The revised manuscript compares against $F_{\text{planning}}$ (incorporating the entropy correction from Lázaro-Gredilla et al., NeurIPS 2024) and Sophisticated Inference (Friston et al., 2021), representing the state-of-the-art in planning-as-inference and active inference respectively. Our scope is planning given a world model, not learning from scratch, so model-free RL addresses a different problem setting.
>
> ## Optimization and scalability evidence
>
> > *"No evidence that the variational formulation is actually easier to optimize or scale in practice."*
>
> The revised Appendix B includes convergence diagnostics across all three environments (5 seeds per configuration), showing reliable convergence with tight confidence bounds. Learning rate and optimization budget sweeps confirm robustness. The MiniGrid experiment provides concrete scalability evidence: temporal factorization reduces parameters from exponential to linear in the horizon, enabling operation where tabular methods cannot function.
>
> ## Stochastic environment and higher-dimensional setting
>
> > *"It would be helpful to include experiments in at least one stochastic environment (to support the plan vs. policy discussion in Section 6.1) and one slightly higher-dimensional or continuous setting to demonstrate that the approach extends beyond tabular cases."*
>
> Both are now addressed: the Reactivity Maze provides the stochastic environment that grounds the plan-versus-policy discussion, and MiniGrid DoorKey-8x8 provides the higher-dimensional setting beyond tabular cases. See the descriptions above.
>
> ## Confidence intervals
>
> All results tables in the revised manuscript now report 95% confidence intervals for proportions and $\pm 1$ standard deviation for continuous values.

---

> > ### Comment · Reviewer_aRw6 · 2026-04-10
> > **Thank you for the revisions**
> >
> > I'd like to thank the authors for additional results reported in the revision, I believe that the paper has been greatly improved. The authors have addressed all concerns raised in my original review.

---

### Decision · Action_Editor_qpMT · 2026-05-10

**Recommendation:** Accept as is

**Audience:**

Yes

**Audience Explanation:**

The paper is a great fit for TMLR and constitutes a nice paper that bridges the gap between two related, but distinct communities.

**Claims And Evidence:**

Yes

**Claims Explanation:**

The paper bridges the gap between the widely used expected free energy principle for planning / decision-making under uncertainty ("acting-as-inference") and variational inference (where a variational free energy is minimized). Algebraically, the connection is relatively straightforward, but the conceptual implications are interesting, and well-discussed and explored in the paper. In a nutshell, the variational free energy can be viewed as a free energy for decision-making where the variational priors play the role of a complexity term which can be viewed as an "epistemic" prior (which needs to be updated in each step; but technically one could decompose this into another "fixed" Bayesian prior and "posterior" pair). This is a nice interpretation and unifies a major principle for acting under uncertainty with a widely used (Bayesian) learning principle. The theory is illustrated with a number of simple experiments.

After the rebuttal and author-reviewer discussion, all reviewers conclude that all claims are supported by accurate, convincing, and clear evidence; and all reviewers recommend acceptance (two accept, one leaning accept). I agree with the reviewers' assessment.